# Position: Supervised Classifiers Answer the Wrong Questions for OOD Detection

Yucen Lily Li [1]   Daohan Lu [1]   Polina Kirichenko [1]   Shikai Qiu [1]   Tim G. J. Rudner [1]   C. Bayan Bruss [2]
Andrew Gordon Wilson [1]

## Abstract

To detect distribution shifts and improve model safety, many out-of-distribution (OOD) detection methods rely on the predictive uncertainty or features of supervised models trained on in-distribution data. In this position paper, we critically re-examine this popular family of OOD detection procedures, and we argue that these methods are fundamentally answering the wrong questions for OOD detection. There is no simple fix to this misalignment, since a classifier trained only on in-distribution classes cannot be expected to identify OOD points; for instance, a cat-dog classifier may confidently misclassify an airplane if it contains features that distinguish cats from dogs, despite generally appearing nothing alike. We find that uncertainty-based methods incorrectly conflate high uncertainty with being OOD, while feature-based methods incorrectly conflate far feature-space distance with being OOD. We show how these pathologies manifest as irreducible errors in OOD detection and identify common settings where these methods are ineffective. Additionally, interventions to improve OOD detection such as feature-logit hybrid methods, scaling of model and data size, epistemic uncertainty representation, and outlier exposure also fail to address this fundamental misalignment in objectives.

## 1 Introduction

In the real world, distribution shifts are the norm rather than the exception. Predictive models are commonly deployed on test data which are drawn from distributions which differ from the training data, such as images acquired from different machines and hospitals, lane boundary detection in different cities, and speech recognition with different accents (Amodei et al., 2016; Jung et al., 2021; Niu et al., 2016; Zhou et al., 2022; Koh et al., 2021). These settings highlight the importance of building models which are robust to natural transformations (Hendrycks and Dietterich, 2018; Mintun et al., 2021; Benton et al., 2020).

However, rather than *generalizing* to natural distribution shifts, it has become popular to *detect* out-of-distribution (OOD) data by training a supervised predictive model on in-distribution data and examining the model's uncertainty, logits, or features. A proliferation of works develop such procedures for detection improvements on known benchmarks (e.g., Hendrycks and Gimpel, 2016; Lee et al., 2018; Ren et al., 2021; Hendrycks et al., 2019a; Liang et al., 2017; Wang et al., 2021), or propose new benchmarks (e.g., Hendrycks et al., 2019a; Yang et al., 2024b; Wang et al., 2022; Bitterwolf et al., 2023; Yang et al., 2021).

Although various limitations of OOD detection methods have been noted, these critiques typically focus on specialized concerns in three distinct categories: (1) issues which motivate minor modifications to existing procedures, such as replacing max softmax with max logit (Hendrycks et al., 2019a), or interventions like Bayesian uncertainty and ensembling (Lakshminarayanan et al., 2017), and outlier exposure (Papadopoulos et al., 2021); (2) criticism of the benchmarks used for detection, such as the conflation of semantic and covariate shift (Yang et al., 2024b)); (3) limitations of specialized models, such inductive biases of coupling layers in normalizing flows (Kirichenko et al., 2020).

By contrast, we do not claim that OOD detection can be solved with better scoring rules, datasets, or models. Instead, **we argue that the entire premise of using supervised models trained on in-distribution data for out-of-distribution detection is fundamentally flawed** — a wholly misspecified enterprise, with no easy fix. We demonstrate that essentially all known OOD detection procedures are answering a fundamentally different question than *"is this point drawn from a different distribution?"*, and so there is a conceptual misalignment which prevents effective OOD detection.

This mismatch becomes especially apparent when detecting the *semantic shifts* associated with previously unseen classes (Yang et al., 2024b). In these cases, supervised models should not be expected to have the necessary information for

[1]New York University [2]Capital One. Correspondence to: Yucen Lily Li <yucenli@nyu.edu>, Andrew Gordon Wilson <andrewgw@cims.nyu.edu>.

*Proceedings of the 42nd International Conference on Machine Learning*, Vancouver, Canada. PMLR 267, 2025. Copyright 2025 by the author(s).

this task. For example, a model trained only to distinguish trucks from cars may confidently misclassify a dog as truck if the features it learned to distinguish cars from trucks also happen to match some features of the dog. It is not that OOD detection is "fundamentally difficult", but rather that current methods answer the wrong question. A dog clearly should be distinguishable from trucks and cars, but a supervised model trained solely on those two categories have no reason to learn this distinction.

In Section 4 we argue in detail that supervised classifiers, trained only to distinguish between in-distribution classes, are misspecified for identifying OOD points (Figure 1). Specifically, we show

- Feature-based methods, which answer *"Does this input lead to features that are far from the features seen during training?"*, cannot reliably identify OOD points. The features for ID and OOD inputs can be very similar, and it is difficult to select only the most discriminative dimensions without access to OOD data. (Section 4.1)

- Uncertainty-based methods answer *"Is the model uncertain about which ID label to assign?"*, but OOD detection requires asking whether the input comes from a different data distribution entirely. As a result, these methods fail in the common scenarios in which ID inputs are inherently ambiguous or OOD inputs are confidently misclassified (Section 4.2).

In Section 5, we then address many of the popular interventions which have been proposed to improve OOD detection, such as using hybrid methods which combine feature and logit-based methods (Section 5.1), encouraging the model to be uncertain on select inputs during training Section 5.2), modeling epistemic uncertainty through Bayesian methods or ensembling (Section 5.3), introducing an "unseen" class during training (Section 5.4), using generative approaches such as normalizing flows and diffusion models (Section 5.5), and scaling up the model and data size (Section 5.6). We find that these approaches all fail to directly address the question of OOD detection, and we identify their conceptual limitations and also illustrate their pathologies through empirical evidence. We conclude in Section 6 with prescriptions based on these findings.

## 2 Preliminaries

**OOD detection task.** The OOD detection task involves identifying points from a different distribution than examples of in-distribution data. Typically, the distribution shifts are categorized as *covariate shifts* and *semantic shifts*. Covariate shifts are *label preserving*, and can include common noise corruptions and transformations, or going from hand-written to type-written characters, or from natural im-

ages to cartoon versions (Hendrycks and Dietterich, 2019; Hendrycks et al., 2021). Semantic shifts involve new unseen classes, where the model has no hope of reaching a correct label. For example, a model could be trained to differentiate between cats and dogs, and then asked to label an airplane. Semantic shift detection is often further categorized into *near OOD*, where the points are similar in some way, and *far OOD* for more distinct inputs. In contrast to OOD detection, *anomaly* and *outlier* detection focus on detecting inputs which significantly differ from other data points (Ruff et al., 2021; Yang et al., 2024a); these inputs could correspond to low-density regions of the training distribution, and do not necessarily indicate that there has been a distribution shift.

**Supervised detection setup.** Let $f_\theta : \mathcal{X} \to \mathcal{Y}$ be a neural network with parameters $\theta$ which maps training data $X^{\mathrm{tr}} \sim p_{\mathcal{X}}(\cdot)$ to class logits. The model's predictive class distribution is given by $p_\theta(y = k|x) = \mathrm{softmax}(f_\theta(x)_k)$ and the final class prediction is $\arg\max_k p_\theta(y = k|x)$. OOD detection methods that leverage trained supervised models propose a scoring function, which for a test example $x^*$, assigns a scalar value $s(x^*, f_\theta, \mathcal{D}_{\mathrm{tr}})$ given a trained model $f_\theta$ and training data $\mathcal{D}_{\mathrm{tr}} = \{X_i^{\mathrm{tr}}, Y_i^{\mathrm{tr}}\}_{i=1}^N$. The score $s(x^*, f_\theta, \mathcal{D}_{\mathrm{tr}})$ is compared to a threshold value to determine whether $x^*$ will be detected as OOD or not. These methods are typically evaluated on distribution shift benchmarks using the ROC (Receiver Operating Characteristic) curve, which plots the true positive rate against the false positive rate at various thresholds, and the AUROC (Area Under the ROC curve), which quantifies the overall performance.

Two particularly common families of approaches have emerged for such OOD detection. If we view the model as a composition of transformations $p_\theta(y = c|x) = \mathrm{softmax}(c_\theta \circ e_\theta(x))_c$ where $e_\theta : \mathcal{X} \to \mathcal{F}$ is the penultimate layer feature extractor, and $c_\theta : \mathcal{F} \to \mathbb{R}^K$ is the classification layer outputting logits, then there are two natural signals to consider — features or logits.

**Feature-based approaches.** These methods compute the OOD score based on the features, typically from the penultimate layer. *Mahalanobis Distance* (Maha) is a common approach (Lee et al., 2018), where we fit a class-conditional Gaussian mixture model to our features with $\mu_c$ and $\Sigma_c$. The final score is computed using the Mahalanobis distance $s(x) = -\min_c (e_\theta(x) - \mu_c)\Sigma^{-1}(e_\theta(x) - \mu_c)^\top$. Many other approaches have also been proposed (Ren et al., 2021; Sun et al., 2022; Tack et al., 2020; Sehwag et al., 2021).

**Logit-based approaches.** These methods operate on the logits of a trained supervised model. The most common approach is *Maximum Softmax Probability* (MSP) (Hendrycks and Gimpel, 2016) $s_{\mathrm{MSP}}(x) = \max_c p_\theta(y = c|x)$. Other popular approaches within this family include the entropy of the predictive distribution $p_\theta(y|x)$ (Ren et al., 2019), value of the maximum logit (Jung et al., 2021) and the energy

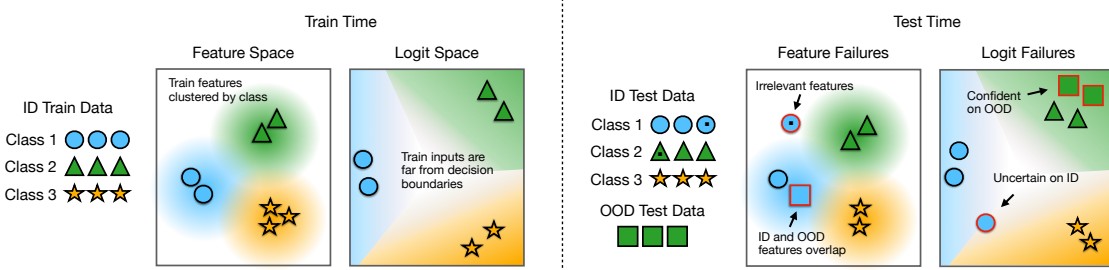

*Figure 1.* **There are irreducible errors when using supervised models for OOD detection because the problem is inherently misspecified.** Supervised models can only determine if an input leads to atypical representations or uncertain predictions, which is fundamentally different than determining if the input belongs to the training distribution. When training on ID data (left), the model accurately clusters the features by class and has high confidence over inputs. However, for OOD detection at test-time (right), the model has distinct failure modes in both the feature and logit-space.

score (Liu et al., 2020). Despite a proliferation of methods, simple approaches such as MSP tend to provide state-of-the-art results, even on the more sophisticated benchmarks (Hendrycks et al., 2019a; Yang et al., 2024b). These methods are thus a natural choice to exemplify the broad conceptual issues with OOD detection, since they provide simple, popular, and still highly competitive approaches.

## 3 Related Work

While anomaly and outlier detection has been studied for many decades in statistics, the related but distinct area of *out-of-distribution* (OOD) detection in deep learning is surprisingly new (Yang et al., 2024a). Amodei et al. (2016) provides a call to action to build methods that are robust to distribution shifts. Shortly after, Hendrycks and Gimpel (2016) proposed using softmax uncertainty as a simple baseline to detect out-of-distribution (OOD) points. A proliferation of methods followed, using the logits, features, or uncertainty of a supervised model trained on in-distribution data to detect out-of-distribution points, achieving better results on benchmark detection tasks (Lee et al., 2018; Liang et al., 2017; Wang et al., 2022; Sun et al., 2022). Other work has focused on introducing new benchmarks with higher resolution images, or test detection, more specifically under *semantic shift* (e.g., new unseen classes) versus covariate shift (label-preserving transformations) (Yang et al., 2024b; Bitterwolf et al., 2023; Huang and Li, 2021). There are also many interventions for boosting performance, including Bayesian uncertainty representation (Lakshminarayanan et al., 2017; Malinin et al., 2019; Tagasovska and Lopez-Paz, 2019; D'Angelo and Fortuin, 2021; Rudner et al., 2022), confidence minimization and outlier exposure (Hendrycks et al., 2018; Papadopoulos et al., 2021; Thulasidasan et al., 2021), and pre-training (Fort et al., 2021; Tran et al., 2022; Hendrycks et al., 2019b).

While there are several works critical in some way of OOD

detection, our focus is significantly different. Critiques tend to be targeted at modifications to existing measures (e.g., max-logit has fewer false positives than MSP) (Hendrycks et al., 2019a), improving the benchmark data (e.g., higher resolution data, data with many classes, and more cleanly separating semantic shift from covariate shift) (Hendrycks and Dietterich, 2019; Yang et al., 2024b), specific architectural properties of generative models (e.g., coupling layers in normalizing flows) (Kirichenko et al., 2020), or note that detection might need to be more tailored to specific shifts (Tajwar et al., 2021; Farquhar and Gal, 2022). By contrast, we examine whether the standard families of methods for OOD detection is *fundamentally misspecified*, answering a different question than "is this point out-of-distribution?"

Another line of OOD detection research relies on deep generative models and likelihoods rather than supervised classifiers. Although generative models have been shown to assign higher likelihood to OOD inputs compared to ID inputs (Hendrycks et al., 2018; Choi et al., 2018; Nalisnick et al., 2019), many alternative methods have been proposed (Ren et al., 2019; Serrà et al., 2019; Morningstar et al., 2021; Graham et al., 2023; Liu et al., 2023). The limitations of these approaches have been studied by Zhang et al. (2021).

## 4 OOD Detection Methods with Supervised Models Answer the Wrong Questions

Many OOD detection methods rely on the features or logits from supervised models that are only exposed to in-distribution data. Even though these approaches are sometimes able to achieve reasonable results on OOD detection benchmarks, they fundamentally answer the wrong question: instead of determining whether an input belongs to the training distribution or some different distribution, they instead ask if the input leads to atypical model representations or unconfident predictions. In this section, we explore the concrete instances where the answers to these two questions

differ, and we demonstrate that feature and logit-based OOD detection methods have irreducible errors as a result.

## 4.1 Feature-Based Methods

Feature-based methods typically use distance metrics to measure how close the features of the test input are to the features of the train inputs, answering the question *"Does this input lead to features that are far from the features seen during training?"*. These methods have two fundamental failure modes: 1) the learned features do not sufficiently discriminate between OOD and ID inputs, and 2) the optimal distance metric depends on the OOD data, forcing these methods to use suboptimal, heuristic-based distance metrics given only access to ID data. In particular, because only a small number of feature dimensions is useful for OOD detection, models typically can not infer these most discriminating features without access to OOD data.

**OOD features can be indistinguishable from ID features.** While OOD inputs generally have unique characteristics that distinguish them from ID data, supervised models may not be incentivized to learn these features if they are unhelpful for ID classification. If the OOD and ID features are indistinguishable, then no feature-based methods can perform well. This failure mode is especially problematic for near OOD detection where fine-grained features are required.

To demonstrate this lack of separability between ID and OOD features, we study four different models trained on ImageNet-1k: ResNet-18, ResNet-50, ViT-S/16, and ViT-B/16, with the OOD datasets of ImageNet-OOD (Yang et al., 2024b), Textures (Cimpoi et al., 2014), and iNaturalist (Van Horn et al., 2018). For each setting, we train an Oracle, a binary linear classifier, to differentiate between examples of ID features and OOD features and report its performance on held-out ID and OOD features. This Oracle serves as a proxy for the best possible performance of any feature-based OOD detection method since it is directly trained on both ID and OOD features, which is not accessible to standard OOD detection methods. We see in Figure 2 (left) that even with ground-truth OOD information, the Oracle is unable to clearly disambiguate between ID and OOD examples on challenging OOD datasets. For each model, $(1-$ Oracle AUROC) represents an irreducible error: no feature-based method can correctly detect these OOD inputs that have indistinguishable features from ID.

**Irrelevant features hurt performance and cannot be fully identified.** Even if the model has learned features which are able to capture the differences between ID and OOD data, it is difficult to separate the relevant distinguishing feature directions from the irrelevant directions. As a result of the underspecification of OOD data at train-time, feature-based methods must rely generic distance metrics in the learned feature space and do not sufficiently up-weight

discriminating features or down-weight irrelevant features.

We demonstrate that feature-based methods are significantly hurt by their inability to perform proper feature selection by comparing them against oracle variants which use only the most distinguishing features. Specifically, we use the features from ResNet-18, ResNet-50, ViT-S/16, and ViT-B/16 trained on ImageNet-1k, and the Mahalanobis (Maha) method, which uses a distance metric defined by the empirical covariance matrix $\Sigma$ of ID features. To determine the optimal subset of distinguishing features, we perform PCA on both ID and OOD features and we retain the number of principal components (chosen from $\{32, 64, 128, 256\}$) which yields the highest Maha AUROC. In Figure 2 (left), we show that using only the most relevant features significantly improves Maha performance on all models by an average of over 10 percentage points. Specifically, the "irrelevant features" error represented in blue represents the difference in AUROC between using Mahalanobis distance on all features compared to using Mahalanobis distance using only the most relevant features for the specific OOD detection task. Moreover, performing this PCA projection accounts for *nearly all* of the reducible error of Maha for the ViT models. In other words, for ViTs, the gap between Maha and the best possible performance is almost entirely explained by the use of irrelevant features in the distance computation, but determining the relevance of features is very challenging without prior access to the OOD data. While methods such as Relative Mahalanobis and ViM (Ren et al., 2021; Wang et al., 2022) use related ideas to attempt to reduce the impact of irrelevant features, they can only use feature covariances computed on ID data alone, and thus do not address this fundamental limitation as we show in Appendix A.1.

In Figure 2 (right), we show that the optimal set of discriminating features is highly specific to the particular OOD dataset we wish to detect and does not transfer. As demonstrated in the first panel using features from ViT-S/16, using the top 32 PCA components computed on ImageNet and IN-OOD improves Maha AUROC in detecting IN-OOD but significantly degrades the AUROC for detecting other OOD datasets like Textures and iNaturalist. This result shows that, as long as the OOD dataset is not specified at training time, determining the influence of irrelevant features is challenging for any feature-based method, presenting another fundamental bottleneck to its detection performance.

**Visual demonstrations.** We visualize clear examples of failure modes for feature-based methods in Appendix A.1. To demonstrate feature overlap, we train a ResNet-18 on a subset of CIFAR-10 classes: airplane, cats, and trucks. We then use this trained model to detect OOD images of dogs. We see in Figure A.1 (left) that the feature space between cats and dog have very overlap, since the model did not learn the features necessary to distinguish between

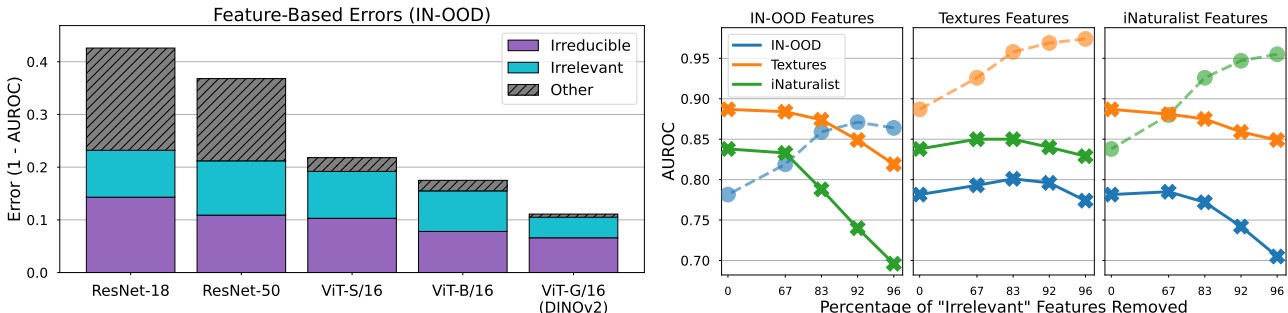

*Figure 2.* **Feature-based methods have two key failure modes: indistinguishable features and irrelevant features. (Left):** The error decomposition of Mahalanobis distance into irrelevant features, indistinguishable features and other components. **(Right):** The most relevant features for OOD detection are specific to each OOD dataset. For ViT-S/16 features, selecting the features that are the most discriminative for one OOD dataset (dashed line) leads to reduced performance on other OOD datasets (solid lines).

these two classes. This pathology is reflected in the poor performance of feature-based methods such as Mahalanobis distance, which only achieves an AUROC of 0.537 and is barely better than random chance. Furthermore, these failures also occur in larger models trained on diverse datasets. Even when using a ResNet-50 trained on ImageNet-1K, Figure A.1 (right) demonstrates that feature-based methods fail to correctly differentiate ID from OOD examples.

## 4.2 Logit-Based Methods

Due to the many pathologies of feature-based OOD detection methods, it may be tempting to instead focus on logit-based methods, which gauge a model's uncertainty over an input's predicted labels via its logits. However, the previous limitations are still applicable. For instance, in the scenario where OOD and ID features overlap, logit-based methods would also fail to detect OOD inputs since the logits are a function of the penultimate-layer features. Furthermore, logit-based methods have their own suite of failure modes which arise from the conflation of *label uncertainty*, the uncertainty over the correct ID label, with *OOD uncertainty*, the uncertainty over whether the sample is ID or OOD. Logit-based methods heuristically assume that higher label uncertainty is equivalent to higher OOD uncertainty, but these are fundamentally different quantities. As a result, there are two distinct failure modes where logit-based methods make the incorrect prediction: instances where ID data naturally has high label uncertainty, and instances where OOD data has low label uncertainty.

**ID examples often have high uncertainty.** To show the misalignment between label and OOD uncertainty, we demonstrate instances where models predict high label uncertainty over in-distribution samples. One example of this failure mode can be found in ImageNet-1K, many of the images within the dataset contain concepts from multiple classes (Stock and Cisse, 2018; Shankar et al., 2020). We would expect these multi-label images to have high label uncertainty since there may be multiple correct answers.

For our experiments, we used the human annotations from Beyer et al. (2020) as the ground truth for the number of labels corresponding to each image.

We explore the behaviors of ResNet-18, ResNet-50, ViT-S/16, ViT-B/16, trained on ImageNet-1k, as well as ViT-G/14 DINOv2 pretrained on internet-scale data. When we apply uncertainty-based metrics to these samples where multiple labels may apply, we find in Figure 3 (left) that the average uncertainty of these multi-label images is significantly higher than corresponding in-distribution samples across a variety of methods. However, these images are clearly ID, since they are sampled from the same distribution that the model was trained on. Furthermore, we see that the logit-based methods are not able to distinguish between ID inputs with high natural label uncertainty and OOD inputs; for example, the AUROC for multi-label images (ID with high label uncertainty) vs ImageNet-OOD is only around 0.6. These results reveal that uncertainty-based methods are insufficient for OOD detection.

**OOD examples often have low uncertainty.** In Figure 3 (right), we consistently find that logit-based approaches are unable to distinguish between ID and the "Striped" class from Textures. Furthermore, in Table A.1, we benchmark 14 different models including ResNets, ViTs, and ConvNext V2 in the setting where ImageNet-1K is ID. We record the FPR@95, which indicates how many OOD examples are incorrectly classified as ID due to their low uncertainty at a threshold where 95% of ID examples are correctly classified. For logit-based methods such as MSP, max-logit, energy score, and entropy, the average FPR@95 across all settings is over 60%; thus, a majority of OOD examples are misclassified due to their low uncertainty.

We provide visual examples of these failure modes of uncertainty-based methods in Appendix A.2, where the predictive uncertainties of ID inputs are indistinguishable from the uncertainties of OOD inputs. In Figure A.4, we note how the uncertainties of an ID and OOD class entirely overlap for

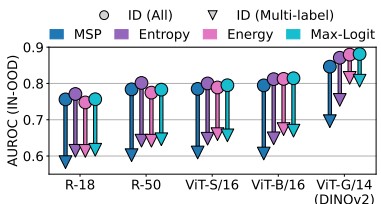 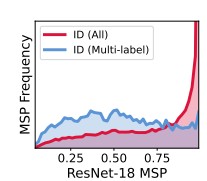 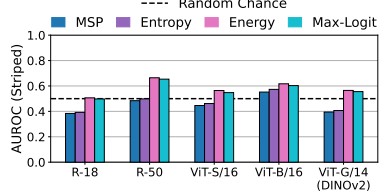 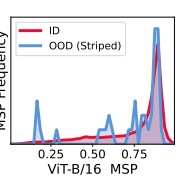

(a) ID inputs with high label uncertainty hurt OOD detection

(b) OOD inputs with low label uncertainty hurt OOD detection

*Figure 3.* **Logit-based methods incorrectly conflate label uncertainty with OOD uncertainty**. **(Left):** The circles show OOD detection on standard ImageNet vs ImageNet-OOD, while the triangles show degraded performance when comparing a subset of ImageNet samples ("Multi-label") against ImageNet-OOD. The histogram shows that the multi-label subset has significantly higher label uncertainty compared to other ID inputs. **(Right):** All methods perform similarly to random chance on the "Striped" class from Textures. The histogram shows that the model's uncertainty on inputs from this OOD class is indistinguishable from its uncertainty on ID samples.

a LeNet-5 trained on a subset of CIFAR-10. We also visualize the feature space of a ResNet-50 trained on ImageNet-1k in Figure A.5 and identify OOD classes which are far from the decision boundary and have high confidence.

Our experiments demonstrate that the difference between label uncertainty and OOD uncertainty, although easy to miss, is a fundamental limitation of logit-based OOD detection methods. This misalignment of goals often leads these methods to exhibit pathological behavior.

## 5 Alternative Views

Given the prevalence of failure modes when using only feature or logit-based OOD methods, numerous strategies have been proposed to enhance OOD performance. In this section, we examine popular interventions such as combining feature and logit-based approaches, pre-training on larger datasets, modeling epistemic uncertainty, and exposing the model to outliers. For these methods, we analyze their limitations, and demonstrate how they fail to address the fundamental pathologies outlined in Section 4. We also address the limitations of explicitly including an OOD class during training and using unsupervised generative models.

### 5.1 Combining Feature and Logit-Based Methods

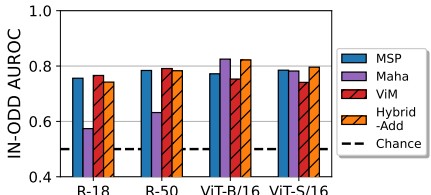

*Figure 4.* Hybrid OOD methods (ViM, Hybrid-Add) do not consistently outperform stand-alone feature-based methods (Maha) and logit-based methods (MSP) and suffer from the same pathologies. Hybrid approaches that combine model features and logits have been proposed for OOD detection (Sun et al., 2021;

Wang et al., 2022), and methods like Virtual-logit Matching (ViM) (Wang et al., 2022) have achieved state-of-the-art results for certain OOD benchmarks. These methods implicitly assume that feature and logit-based methods have complementary failure modes.

To probe this assumption, we introduce a simple baseline method, "Hybrid-Add", which sums the normalized scores of a feature-based method (Mahalanobis) with a logit-based method (MSP). We find that Hybrid-Add improves OOD detection for some models compared to using MSP or Mahalanobis alone on Textures (Figure A.8), indicating feature and logit-based methods can have distinct failure modes. However, this benefit is highly model and dataset-specific; on other OOD datasets like iNaturalist and IN-OOD, Hybrid-Add does not offer a clear advantage over MSP or Mahalanobis, as seen in Figure 4 and Figure A.8. Other hybrid methods such as ViM also do not consistently outperform MSP and Mahalanobis distance. These findings demonstrate that feature and logit-based methods do not always share distinct failure modes.

Crucially, hybrid methods do not resolve the fundamental pathologies of OOD detection due to model misspecification. In the many cases where ID and OOD features are indistinguishable, as discussed in Section 4.1, both logit and features do not provide reliable information for OOD detection. Furthermore, hybrid methods do not circumvent any of the previously identified pathologies of feature or logit-based methods. Therefore, although they may offer improvements on select benchmarks, hybrid methods do not address the fundamental bottlenecks.

### 5.2 Exposing to Outliers

Another popular approach to improve OOD detection is outlier exposure, which incorporates OOD examples when training the model (Hendrycks et al., 2018; Choi et al., 2023). In this setting, we explicitly optimize the model to have high uncertainty on the outlier dataset:

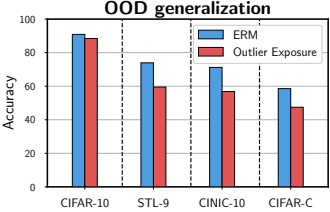

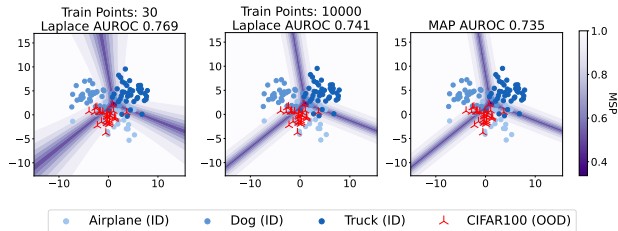

*Figure 5.* Training a ResNet-18 with outlier exposure hurts OOD generalization for covariate shifts compared to standard training.

*Figure 6.* Epistemic uncertainty becomes less useful for OOD detection as more ID data is observed due to posterior collapse.

$$\mathcal{L} = \mathbb{E}_{(x,y)\sim\mathcal{D}_{\text{in}}}\ell_{\text{CE}}(f(x),y) + \alpha\mathbb{E}_{x'\sim\mathcal{D}_{\text{out}}}\ell_{\text{CE}}(f(x'),y_u)$$
where $y_u$ is the uniform distribution over $K$ classes. Outlier exposure relies on the dataset $\mathcal{D}_{\text{out}}$ to encourage the model to have high predictive uncertainty away from the training data. However, even if the model is exposed to OOD data during training, the final model is still misspecified because it only contains ID classes as possible categorizations. As discussed in Section 4.1, OOD datasets are quite diverse, and the features necessary to distinguish ID from one OOD dataset often do not generalize.

Furthermore, *outlier exposure may significantly hurt OOD generalization* because the model is explicitly trained to have high label uncertainty over a large set of inputs; this degradation in performance is especially problematic because OOD generalization is essential for model robustness and reliability. To demonstrate this behavior, we compare two ResNet-18 models trained on CIFAR-10, one with the standard training regime and the other with outlier exposure using TIN-597 as $\mathcal{D}_{\text{out}}$ following Zhang et al. (2023) (see Appendix B.1 for setup details).

In Appendix A.7, we show that outlier exposure does improve OOD detection for most of the semantic shift OOD benchmarks. However, outlier exposure does not improve performance on MNIST, likely because this dataset differs significantly from natural image benchmarks. This decreased performance highlights the sensitivity of outlier exposure to the choice of OOD data and reiterates that the features which distinguish ID and OOD are not consistent across diverse OOD datasets. Furthermore, we find that both the ID accuracy and the OOD generalization of the model are significantly negatively impacted. In Figure 5, on inputs with covariate shifts, outlier exposure hurts the model's accuracy by over 10% across our benchmarked datasets. Thus, by explicitly encouraging high uncertainty on the diverse outlier dataset, we sacrifice our model's generalizability.

### 5.3 Modeling Epistemic Uncertainty

Predictive uncertainty can be separated into *aleatoric uncertainty*, which is considered irreducible and stems from inherent data variability, and *epistemic uncertainty*, which is uncertainty over which solution is correct given the lim-

ited data. It has been posited that focusing on epistemic uncertainty is the principled approach to OOD detection because the uncertainty increases as we move away from the data, and there is a proliferation of methods approximating epistemic uncertainty for improved OOD detection (e.g., Band et al., 2021; D'Angelo and Fortuin, 2021; Lakshminarayanan et al., 2017; Malinin et al., 2019; Rudner et al., 2022; Tagasovska and Lopez-Paz, 2019; Tran et al., 2022).

Epistemic uncertainty is typically represented through a distribution over the model parameters. For a model $f$ with stochastic parameters $\Theta$, distributed according to $q(\theta)$, we can express the model's predictive uncertainty as

$$\underbrace{\mathcal{H}\left(\mathbb{E}_{q_\Theta}[p(y\mid\mathbf{x},\Theta)]\right)}_{\text{Predictive Unc.}} = \underbrace{\mathbb{E}_{q_\Theta}[\mathcal{H}(p(y\mid\mathbf{x},\Theta))]}_{\text{Aleatoric Unc.}} + \underbrace{\mathcal{I}(Y;\Theta)}_{\text{Epistemic Unc.}}$$

where $\mathcal{H}(\cdot)$ is the entropy functional and $\mathcal{I}(Y;\Theta)$ is the mutual information. The predictive distribution is then $p(y=c|\mathbf{x},\mathcal{D}) = \int \text{softmax}(f_\theta(\mathbf{x}))_c \cdot p(\theta|\mathcal{D})d\theta$.

We note that *deep ensembling* (Lakshminarayanan et al., 2017), popular for OOD detection, is a prominent example of epistemic uncertainty representation; by marginalizing over modes in a posterior they often provide a relatively accurate representation of the posterior predictive distribution (Wilson and Izmailov, 2020; Izmailov et al., 2021).

However, the predictive uncertainty is not over whether a point is OOD but rather over class labels, and epistemic uncertainty does not address this limitation. Consider how epistemic uncertainty changes as a function of data size. In the infinite ID-data limit, the posterior over model's parameters collapses, and the model becomes extremely confident in its parameters. If measuring epistemic uncertainty were the correct approach to OOD detection, then low epistemic uncertainty implies that OOD points do not exist in this setting. Therefore, because perfectly capturing epistemic uncertainty is not enough to solve OOD detection, they must answer fundamentally different questions.

To illustrate this phenomenon, we consider a last-layer Bayesian approximation (Kristiadi et al., 2020) and train a linear layer $f_\theta$ over features extracted from a ResNet-18 trained on IN-1K to classify three classes: airplane, dog, and

truck. We place a prior over parameters $\theta$, and we approximate the predictive distribution through a Laplace approximation that uses a Gaussian distribution to approximate the posterior distribution of the model parameters, allowing for the estimation of epistemic uncertainty (MacKay, 2003). In Figure 6, we visualize the learned decision boundaries by applying PCA to reduce the logit space to two dimensions and plot the MSP over this projection. As the size of the training data increases, the posterior noticeably contracts and the performance worsens, approaching that of a deterministic model using maximum a posteriori (MAP) estimates!

## 5.4 Introducing an Unseen Class

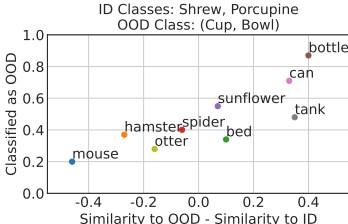

Figure 7. Adding an OOD class is only effective if the train OOD examples are similar to test OOD examples.

Since standard classification models trained over $K$ classes are fundamentally misspecified for the task of detecting OOD classes in both feature-space and logit-space, it may be tempting to correct the specification problem by adding a $(K + 1)$-th class corresponding to the OOD category (Thulasidasan et al., 2020). During training, we can then expose models to OOD examples, and use this additional class for OOD detection on test samples.

However, we find this method is only effective when the examples that the model is exposed to during training are very similar to the OOD examples during test-time, which is often unrealistic. To demonstrate, we train a ResNet-18 model on two CIFAR-100 classes: keyboard and porcupine, and use samples from cup and skyscraper for the OOD class. We then measure the performance of OOD detection over the remaining CIFAR-100 classes. In Figure 7, we use BERT embeddings (Devlin et al., 2018) to compute the cosine similarity of the test-time OOD classes to the train-time ID and OOD classes. We see that the OOD class is effective for test-time examples of bottle and can, since they are similar to the train OOD examples. However, the model is unable to accurately categorize examples like hamster and mouse, which are more closely related to ID classes.

## 5.5 Generative Models

Unlike supervised classifiers, unsupervised generative models trained on the ID dataset directly measure the likelihood of sample $x$ under the training distribution. Generative models, therefore, may appear to be a principled solution to

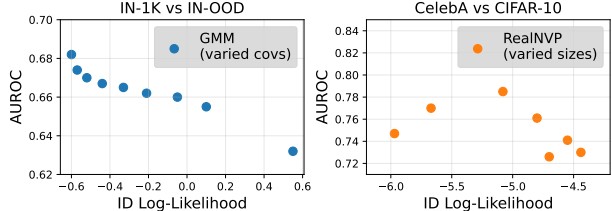

Figure 8. **Better generative models of ID data can lead to worse OOD detection**. **Left:** We create GMMs for ResNet-50 ImageNet features with various covariance matrices ranging from the empirical covariance estimations to identity matrix. The models which better fit the ID data have worse OOD detection. **Right:** We benchmark RealNVP models of different sizes, and find the models which achieve better likelihoods on ID CelebA images do not consistently achieve better AUROC for detecting CIFAR-10.

OOD detection. However, better generative models are not always better OOD detectors: $p(x)$ answers a fundamentally different question than $p(\text{OOD}|x)$, and there is generally a conflict between creating a better model for $p(x)$ and the ability to use the likelihoods for OOD detection.

In Figure 8 (left), we demonstrate this conflict by constructing Gaussian Mixture Model (GMM) models for the ImageNet-1K features from a ResNet-50. We create a GMM with one cluster per class, using class-conditional means and covariances estimated from training data. This model, equivalent to the generative model used by the Mahalanobis method (Lee et al., 2018), achieves high likelihood on ID data (seen as the rightmost dot). However, as we degrade the generative model by interpolating the covariance towards the trivial identity covariance, the OOD detection improves monotonically. We further illustrate this phenomena in Figure 8 (right), where we show the performance of RealNVP normalizing flow models (Dinh et al., 2016) of various sizes trained on CelebA images. Again, we find that the quality of the generative model does not have a strong association with its ability to detect OOD images from CIFAR-10.

We further demonstrate this misalignment in Appendix A.6, discussing additional limitations and illustrating the failures of generative approaches such as diffusion models.

## 5.6 Scaling Model and Data Size

Increasing model size and pre-training on large datasets have been shown to reliably improve OOD detection benchmarks as models tend to learn more diverse and higher-quality features (Fort et al., 2021; Dehghani et al., 2023; Miyai et al., 2023). When models see more diverse data and as the model capacity increases, they can learn more features that help distinguish OOD and ID data.

However, as we show in Figure 9 and Figure A.7, scaling

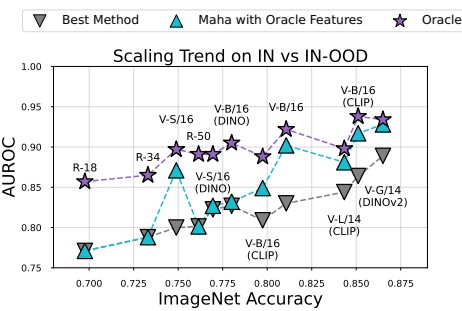

*Figure 9.* Even scaling to ViT-G/14 DINOv2 pre-trained on internet scale data (right-most), the best method still contains significant error from indistinguishable features and irrelevant features.

alone does not fully address the limitations of OOD detection methods. We benchmark twelve different models of varying sizes and pretraining methods, enumerated in Appendix B.3. In challenging near-OOD problems such as ImageNet vs. ImageNet-OOD, models learn additional discriminating features between ID and OOD data at an extremely slow rate, such that even the largest ViT-G/14 DINOv2 still has over $5\%$ irreducible error due to indistinguishable features. As we have argued in Section 4, this error can not be decreased regardless of what OOD detection method we use. Indeed, the AUROC achieved by the best method (Best) among Maha, Rel Maha, MSP, Max Logit, Energy, and ViM is consistently below the Oracle binary classifier trained on ID and OOD features. Furthermore, while the error due to indistinguishable features may be decreased (slowly) with scale, there is still a large gap between the best existing method and the Oracle as we scale the model. Much of this gap can be recovered by optimally selecting features for Maha (Maha with Oracle Features), suggesting that the presence of irrelevant features continues to limit the performance. We provide additional empirical results in Appendix A.5 which demonstrate the scaling behaviors using 54 models over nine architectures and six pre-training setups. These results demonstrate the fundamental pathologies of existing OOD detection methods even with increasing model and data size.

## 6 What Should We Do?

Moving forward, we must be intentional about choosing between the fundamentally different goals of OOD generalization vs detection. Many real-world use-cases may prefer graceful generalization under mild covariate shifts rather than simply detection. For example, we may want models to handle noisy images or adapt from hand-written to type-written digits rather than merely flag these inputs as different. In these cases, interventions such as outlier exposure, which train the model to be uncertain for OOD points, often cause the model to generalize *more poorly*

under covariate shifts and lead to counterproductive results.

If we are interested in OOD detection rather than generalization, we must distinguish between semantic and covariate shift detection. As we have demonstrated, semantic shift detection is often not a sensible objective with standard procedures which use the features or logits of a supervised model. Hybrid OOD detection methods which combine feature-space and logit-space information also do not address the fundamental pathologies caused by the problem misspecification and thus fail to reliably perform well.

Other interventions which estimate epistemic uncertainty, such as Bayesian methods and ensembles, are also fundamentally misaligned with the OOD detection objective, and become *worse* at distinguishing between in-distribution and OOD points as we acquire more in-distribution data — exactly the opposite behavior we would desire! We should use these methods with caution, especially in cases where we have access to significant amounts of in-distribution data.

The most natural way to resolve the model misspecification of supervised procedures is to introduce a new class representing points from other distributions (e.g., a three-class classifier for 'cats', 'dogs', and 'anything else'). However, for this approach to work, the examples from the 'anything else' class have to share common structure and should be carefully selected to be representative of the types of OOD inputs we want to detect at test-time.

Unsupervised methods, such generative models or density estimation, also suffer from fundamental limitations. $p(x)$ answers a different question than $p(\text{OOD}|x)$, creating a tension between the quality of density estimation and performance in OOD detection. Moreover, we are more interested in typicality than density (regions with high density can have low mass, meaning they would not be typical samples), but different notions of typicality can lead to very different OOD detection behavior, and choosing amongst these notions can be arbitrary.

If we are to use a supervised procedure for OOD detection, extensive pre-training over many classes can be a useful intervention and generally leads to improved performance across various OOD detection methods. However, scaling the model size and pre-training data is insufficient for addressing the misaligned objectives, and even models like ViT-G/14, pretrained on internet-scale data using self-supervised objectives, suffer pathological behavior.

Ultimately, effective out-of-distribution detection requires methods that directly address the core question rather than relying on convenient heuristics. Rather than continuing to utilize techniques which are fundamentally misaligned with the goal of OOD detection, we should instead develop principled approaches that directly estimate the probability that an input comes from a different distribution.

## Acknowledgments

We thank Marc Finzi, Micah Goldblum, Sanae Lotfi, Nate Gruver, and Sanyam Kapoor for helpful discussions and thought-provoking questions. This work was supported by NSF CAREER IIS-2145492, NSF CDS&E-MSS 2134216, NSF HDR-2118310, BigHat Biosciences, Capital One, and the NYU IT High Performance Computing resources, services, and staff expertise.

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

# A  Additional Empirical Results

## A.1  Feature-based methods

Feature-based methods have two distinct failure modes: indistinguishable features and irrelevant features. In this section, we provide further empirical demonstrations of these failure modes across various models and datasets.

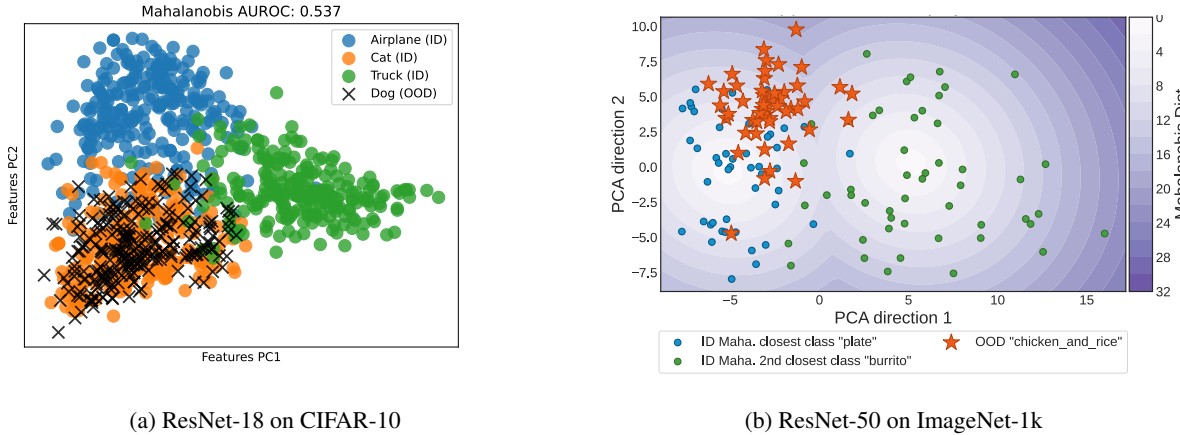

(a) ResNet-18 on CIFAR-10                 (b) ResNet-50 on ImageNet-1k

*Figure A.1.* Visualizations of failure modes for feature-based OOD detection. **(Left):** We train a ResNet-18 on a subset of CIFAR-10, and find that the feature space between an ID class and OOD class have significant overlap. **(Right):** Feature-based methods also fail for larger models like ResNet-50 trained on ImageNet 1K, where OOD classes have low Mahalanobis distance.

To show that the features that a model learns to distinguish ID data may be insufficient for OOD detection, we train a ResNet-18 on a subset of CIFAR-10 with only three classes: airplanes, cats, and trucks, and we achieve a test loss of 0.95 accuracy, indicating that our model is able to distinguish between the three ID classes. However, when we introduce the OOD class of dog, we can see in Figure A.1 (left) that the features for dog almost entirely overlap with the features for cat. We visualize the features by doing PCA and plotting the first two dimensions. Because of this overlap, the feature-based Mahalanobis distance only achieves an AUROC of 0.537, which is very close to random chance.

We find similar behaviors for larger models and datasets. In Figure A.1 (right), we visualize the features of various for a ResNet-50 trained on ImageNet-1k. We see that the ID classes, represented in blue and green dots, are fairly distinct in the feature space. However, the OOD class has significant overlap with the blue points, and the model is unable to clearly disambiguate between ID and OOD points by only looking at the Mahalanobis distance.

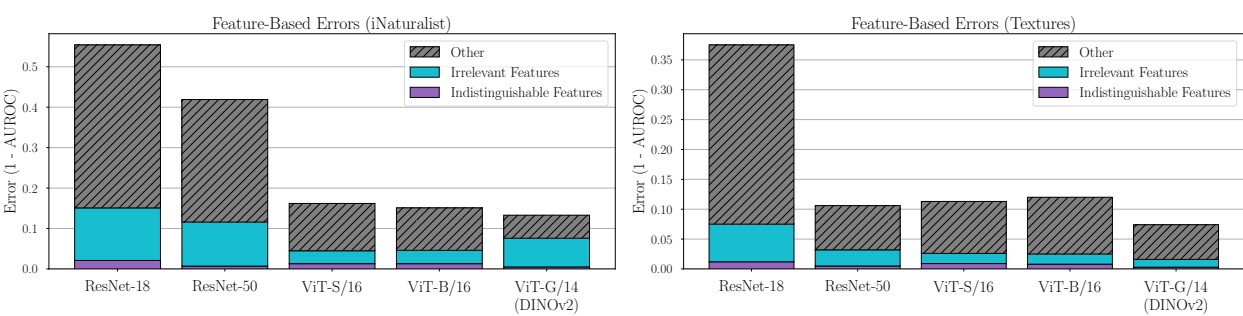

*Figure A.2.* Error decomposition of feature-based methods

We provide error decompositions of these failure modes across various OOD datasets in Figure A.2, where we follow the same experiment setup as in Figure 2 (left): we use the features from ResNet-18, ResNet-50, ViT-S/16, and ViT-B/16 trained on ImageNet-1k, and the Mahalanobis (Maha) method, which uses a distance metric defined by the empirical covariance matrix $\Sigma$ of ID features. To determine the optimal subset of distinguishing features, we perform PCA on both ID and OOD features and we retain the number of principal components (chosen from $\{32, 64, 128, 256\}$) which yields the highest

Mahalanobis AUROC. We refer to this optimal score as the "Oracle PCA" because it utilizes the true ID and OOD features, which is not possible in realistic settings. In Figure A.2, we see that the a significant amount of the error for iNaturalist and Textures come from irrelevant features, and the error could be greatly reduced if we had an understanding of which feature dimensions would be the most relevant for OOD detection. However, because the optimal features is heavily dependent on the OOD dataset as shown in Figure 2 (right), this information cannot be determined when we only have access to ID data.

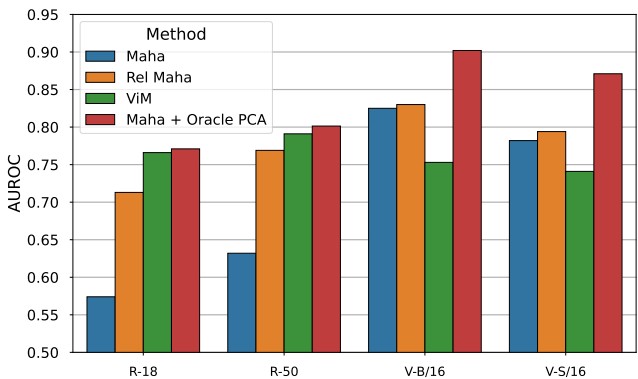

*Figure A.3.* **Relative Mahalanobis and ViM do not fully address the issue of irrelevant features on ImageNet vs ImageNet-OOD, especially with the more performant ViT models.** In all cases, Mahalanobis with an Oracle PCA performs the best. Except for ResNets, Relative Mahalanobis and ViM offer negligible or negative improvement relative to Mahalanobis. The gap between Maha + Oracle PCA and the best-performing feature-based method is especially large for ViTs.

We demonstrate that advanced feature-based methods like Relative Mahalanobis, which uses heuristics to identify some irrelevant features, and ViM, which combines features with logits, are subject to the same failure modes in Figure A.3. We find that for ViT models, the performance of Relative Mahalanobis or ViM is negligble or even negative compared to standard Mahalanobis distance. However, when we look at the red bar, which demonstrates the best Mahalanobis distance when using an oracle feature selection process, we see that this method outperforms both Relative Mahalanobis and ViM. This suggests that the irrelevant features are a major problem for all feature-based methods, even hybrid ones, and this problem is not easily resolved when we only have access to ID datasets.

## A.2 Logit-based methods

Logit-based methods fail when the uncertainty of ID data looks similar to the uncertainty of OOD data.

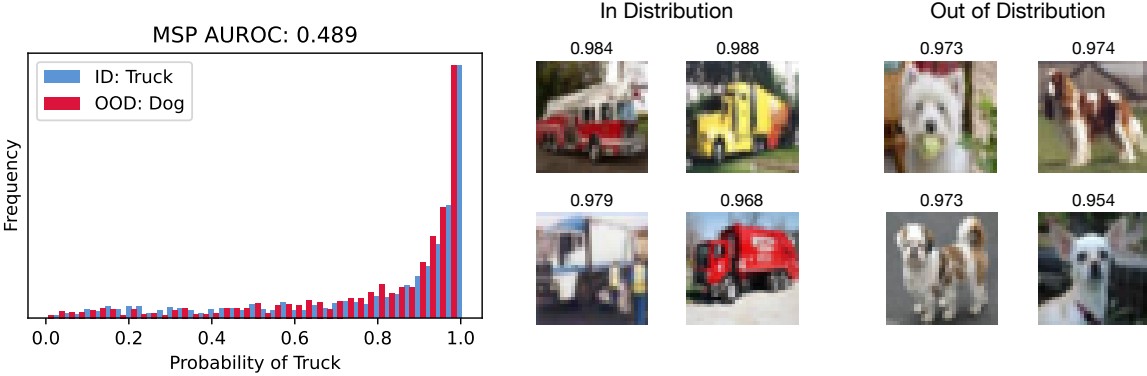

(a) Confidence of ID vs OOD inputs          (b) Example inputs and model confidence (MSP)

*Figure A.4.* **The predictive uncertainty of OOD points may be indistinguishable from ID points.** We train a LeNet5 to classify CIFAR10 automobiles and trucks, and we test the OOD dog class. We see that the model confidence for OOD dogs entirely overlaps with the ID truck class. In this setting, because the uncertainties are identical, no uncertainty-based method would be able to successfully differentiate ID from OOD.

To demonstrate this failure mode, we construct a simple example where we train a LeNet 5 to classify CIFAR10 automobiles and trucks, and we look at how the model performs on the OOD class od dogs. In Figure A.4, where we find that the model very confidently classifies OOD dogs as ID trucks. In the left size of the figure, we see that the model's predictive uncertainty for the ID class "Trucks" is nearly identical to the uncertainty for the OOD class "Dogs", indicating that uncertainty-based methods will not be effective for OOD detection. We see in the Figure A.4 (right) that these high confidences are not due to label noise; the model very confidently classifies images of what are clearly dogs as trucks.

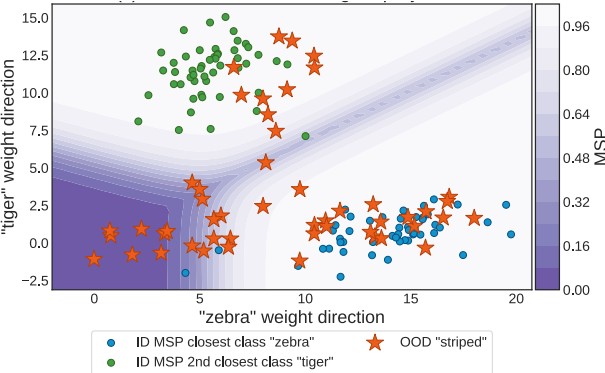

*Figure A.5.* For a ResNet-50 trained on ImageNet-1k, we see that the model has very high confidence for the OOD class 'Striped', highlighting the difference between label uncertainty and OOD uncertainty.

We also demonstrate similar failure modes on larger models. Figure A.5 visualizes the uncertainties for a ResNet-50 trained on ImageNet. We see that the model often has high confidence for many instances of the OOD class 'Stripes', and so this OOD class is incorrectly classified as OOD. In Figure A.6, we find that this failure mode is prevalent both ResNets and ViTs, and for many logit-based methods such as MSP, Max-Logit, Energy, and Entropy. Therefore, this demonstrates a fundamental pathology where the many models are consistently overconfident on OOD inputs.

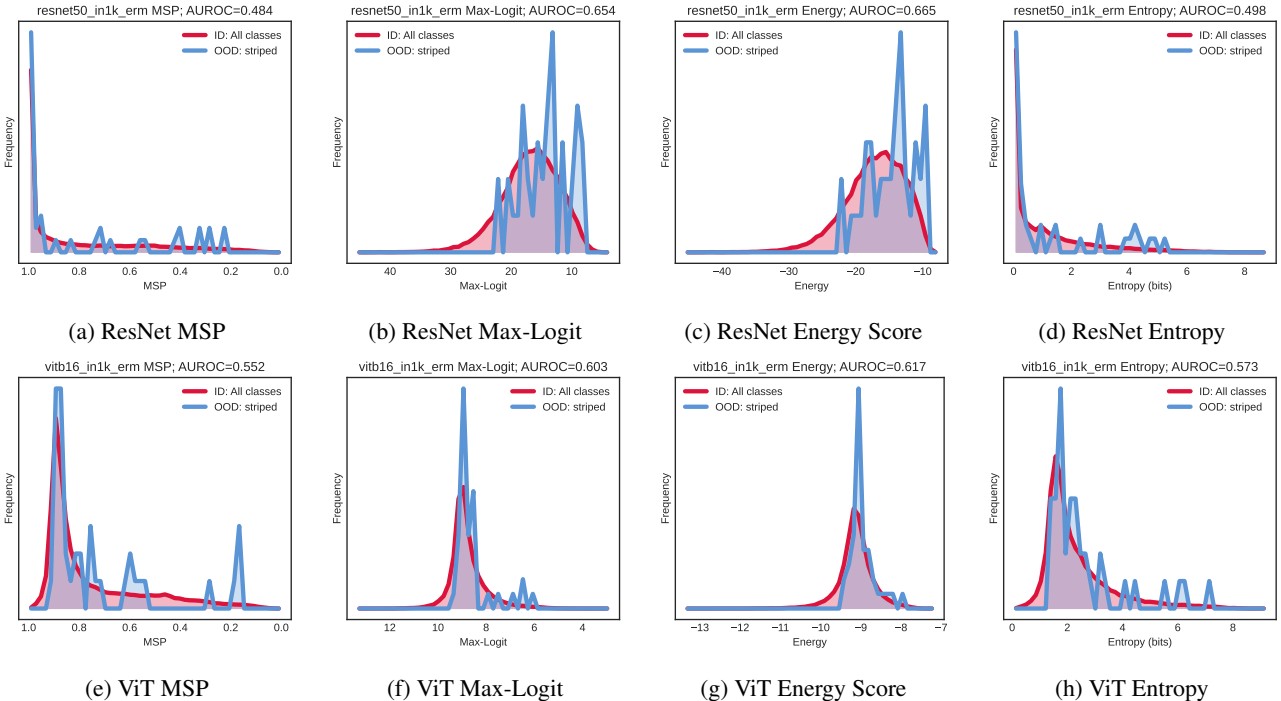

(a) ResNet MSP    (b) ResNet Max-Logit    (c) ResNet Energy Score    (d) ResNet Entropy

(e) ViT MSP    (f) ViT Max-Logit    (g) ViT Energy Score    (h) ViT Entropy

*Figure A.6.* We plot the distribution of OOD scores on ImageNet-1K ID and Describable Textures 'striped' class OOD data obtained from different OOD detection methods. We discover a systematic failure mode of all methods that utilize logits stemming from the model being overconfident about its predictions on the OOD data. Even though different OOD detection methods have different AUROC numbers, the score distribution plots reveal it is difficult to cleanly separate ID and OOD scores by picking a threshold. We use a ResNet-50 pretrained on ImageNet-1K and use a ViT-B/16 pretrained on ImageNet-1K.

In Table A.1, we record the FPR@95, which indicates how many OOD examples would be classified as ID at a threshold where 95% of the ID examples are correctly classified. We find that well over half of OOD examples are misclassified as ID even with powerful pre-trained models, demonstrating that logit-based methods as they are currently set up are unable to accurately differentiate ID from OOD inputs.

| OOD Dataset | Model | MSP | Max Logit | Entropy | Energy Score |
|---|---|---|---|---|---|
| IN-OOD | ResNet-50 | 0.774 | 0.804 | 0.820 | 0.778 |
| IN-OOD | ResNet-50 DINO | 0.804 | 0.830 | 0.847 | 0.823 |
| IN-OOD | ResNet-34 | 0.807 | 0.824 | 0.838 | 0.809 |
| IN-OOD | ResNet-18 | 0.832 | 0.846 | 0.855 | 0.845 |
| IN-OOD | ViT-S/16 | 0.797 | 0.803 | 0.818 | 0.798 |
| IN-OOD | ViT-S/16 DINO | 0.761 | 0.790 | 0.811 | 0.768 |
| IN-OOD | ViT-B/16 | 0.740 | 0.733 | 0.771 | 0.726 |
| IN-OOD | ViT-B/16 DINO | 0.741 | 0.764 | 0.784 | 0.749 |
| IN-OOD | ViT-B/16 CLIP | 0.764 | 0.776 | 0.805 | 0.726 |
| IN-OOD | ViT-B/14 DINOv2 | 0.658 | 0.621 | 0.638 | 0.610 |
| IN-OOD | ViT-G/14 DINOv2 | 0.562 | 0.448 | 0.450 | 0.469 |
| IN-OOD | ViT-L/14 CLIP | 0.686 | 0.685 | 0.723 | 0.631 |
| IN-OOD | ConvNeXt V2-B | 0.701 | 0.708 | 0.773 | 0.673 |
| IN-OOD | ConvNeXt V2-L | 0.696 | 0.710 | 0.787 | 0.663 |
| Textures | ResNet-50 | 0.662 | 0.544 | 0.522 | 0.594 |
| Textures | ResNet-50 DINO | 0.681 | 0.624 | 0.612 | 0.637 |
| Textures | ResNet-34 | 0.690 | 0.562 | 0.533 | 0.620 |
| Textures | ResNet-18 | 0.710 | 0.571 | 0.527 | 0.643 |
| Textures | ViT-S/16 | 0.672 | 0.579 | 0.506 | 0.593 |
| Textures | ViT-S/16 DINO | 0.612 | 0.400 | 0.363 | 0.521 |
| Textures | ViT-B/16 | 0.586 | 0.544 | 0.573 | 0.521 |
| Textures | ViT-B/16 DINO | 0.531 | 0.351 | 0.307 | 0.437 |
| Textures | ViT-B/16 CLIP | 0.657 | 0.530 | 0.538 | 0.564 |
| Textures | ViT-B/14 DINOv2 | 0.535 | 0.409 | 0.401 | 0.451 |
| Textures | ViT-G/14 DINOv2 | 0.480 | 0.344 | 0.332 | 0.389 |
| Textures | ViT-L/14 CLIP | 0.543 | 0.441 | 0.446 | 0.462 |
| Textures | ConvNeXt V2-B | 0.530 | 0.480 | 0.490 | 0.441 |
| Textures | ConvNeXt V2-L | 0.551 | 0.468 | 0.480 | 0.440 |
| iNaturalist | ResNet-50 | 0.703 | 0.700 | 0.716 | 0.684 |
| iNaturalist | ResNet-50 DINO | 0.644 | 0.594 | 0.598 | 0.619 |
| iNaturalist | ResNet-34 | 0.745 | 0.721 | 0.726 | 0.728 |
| iNaturalist | ResNet-18 | 0.739 | 0.727 | 0.734 | 0.727 |
| iNaturalist | ViT-S/16 | 0.726 | 0.683 | 0.668 | 0.692 |
| iNaturalist | ViT-S/16 DINO | 0.726 | 0.660 | 0.658 | 0.699 |
| iNaturalist | ViT-B/16 | 0.692 | 0.711 | 0.791 | 0.674 |
| iNaturalist | ViT-B/16 DINO | 0.682 | 0.617 | 0.613 | 0.648 |
| iNaturalist | ViT-B/16 CLIP | 0.698 | 0.655 | 0.683 | 0.634 |
| iNaturalist | ViT-B/14 DINOv2 | 0.519 | 0.429 | 0.426 | 0.455 |
| iNaturalist | ViT-G/14 DINOv2 | 0.448 | 0.355 | 0.351 | 0.384 |
| iNaturalist | ViT-L/14 CLIP | 0.593 | 0.547 | 0.566 | 0.522 |
| iNaturalist | ConvNeXt V2-B | 0.634 | 0.638 | 0.712 | 0.589 |
| iNaturalist | ConvNeXt V2-L | 0.627 | 0.624 | 0.704 | 0.563 |

*Table A.1.* **FPR@95 for OOD detection remains high with popular models**. We record the FPR@95 for the MSP method for 14 models including ResNets, ViTs, and ConvNext V2 models on ImageNet-1K as ID, and Textures, iNaturalist, and ImageNet-OOD as OOD. FPR@95 records how many OOD examples are classified as ID (low uncertainty, false positive) at a threshold where 95% of ID examples are correctly classified (true positive). The average FPR@95 over all models and OOD datasets is 66.5%, thus well over half of OOD examples are classified as ID due to having low uncertainty, and other methods such as max logit, energy score, and entropy all have similar FPR@95s of over 60%.

### A.3  Scaling

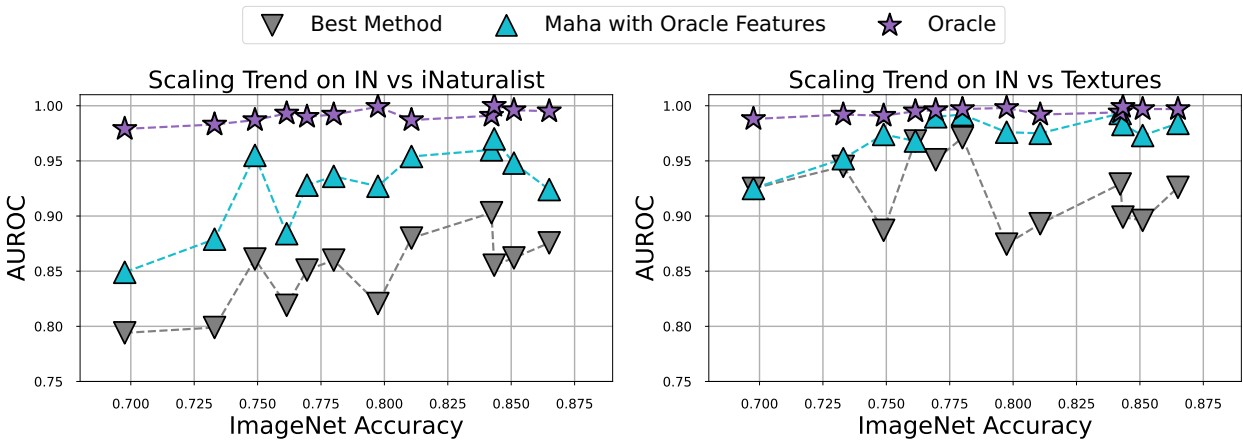

*Figure A.7.* Scaling trends for ImageNet as ID and iNaturalist and Textures as OOD

We benchmark the failure modes for 12 different models, detailed in Appendix B.3. For each model, we perform PCA across all ID and OOD features, and we keep the number of principal components (chosen from $\{32, 64, 128, 256\}$) which yields the highest Mahalanobis AUROC. We refer to this optimal score as the "Maha with Oracle Features" because it utilizes the true ID and OOD features, which is not possible in realistic settings.

We see in Figure A.7 that even large models trained on internet-scale data still contain the pathologies of indistinguishable features and irrelevant features. The gap between the "Best Method" and "Maha with Oracle Features" represents the failure mode of irrelevant features, where the methods are unable to distinguish between the dimensions which are useful or not useful for OOD detection. For both the iNaturalist as well as the Textures OOD dataset, we find that the Oracle Features consistently outperforms the best method by a significant margin, indicating that the irrelevant features are hurting the performance of these methods.

### A.4  Hybrid Methods

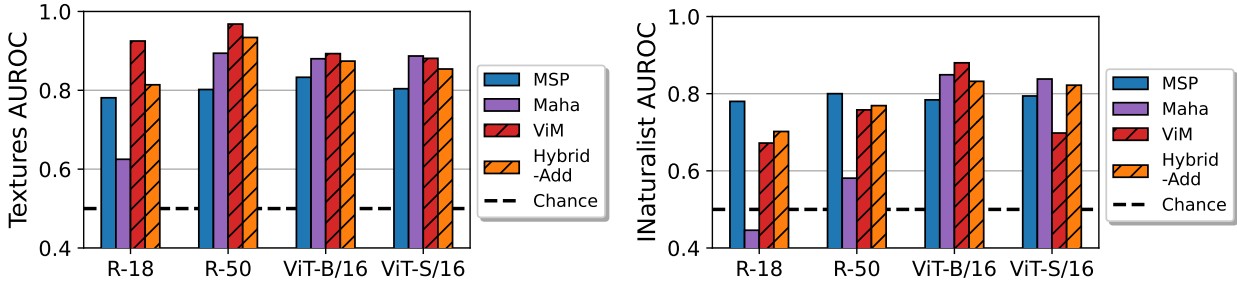

*Figure A.8.* Hybrid OOD methods outperform logit and feature-based on Textures

We find hybrid methods like ViM and Hybrid-Add work well on far-OOD datasets like Textures, where we see noticeable improvement across many models in Figure A.8.

## A.5 Effect of Pre-training

Miller et al. (2021) showed that there is a strong linear relationship between ID accuracy and OOD generalization on OOD data with covariate shifts, suggesting it is sufficient to focus on improving ID accuracy for better robustness. Similarly, we explore the connection between the test accuracy and OOD detection performance. We use ImageNet-1K (IN-1K) as ID data and ImageNet-OOD (IN-OOD) (Yang et al., 2024b) and Textures (Cimpoi et al., 2014) as OOD data. We evaluate 54 models covering a wide range of architectures and pretraining methods. In Figure A.9 we plot AUROC of MSP against ImageNet test accuracy. Generally, ID accuracy and AUROC have close to a linear relationship for models with low- to medium-range performance on ImageNet. However, on ImageNet-OOD for models performing around or better than 75% ID accuracy, we observe higher variability in AUROC: for larger-scale highly performant models pre-training data impacts the OOD detection more significantly.

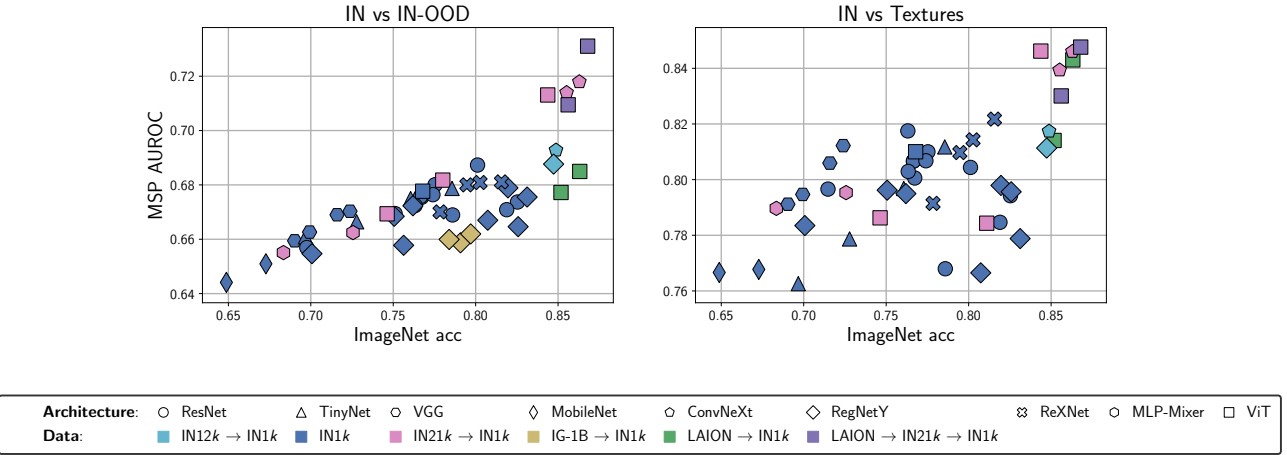

*Figure A.9.* The impact of the model architecture, pre-training data and objective on OOD detection performance. AUROC of MSP on ImageNet vs ImageNet-OOD (left) and ImageNet vs Textures (right) against ImageNet test accuracy. We observe that improving ImageNet accuracy generally leads to better OOD detection.

When models are exposed to a diverse set of data during pre-training, they are likely to learn a wide range of features, making it possible for them to differentiate between ID and semantically new classes. There is an edge case when the OOD data is included in pre-training dataset: in Figure A.9 for the best performing ViT and ConvNext models pre-training includes ImageNet-21K, after which they are fine-tuned on ImageNet-1K. Since ImageNet-OOD consists of images from ImageNet-21K which do not semantically overlap with ImageNet-1K classes, we observe a rapid jump in AUROC for these models with negligible variability in ID accuracy. Pre-training on diverse data which includes similar examples to OOD points softens the misspecification of the MSP approach and leads to strong performance.

## A.6 Generative models

**Conceptual limitation of generative models for OOD detection.** Estimating $p(x)$ is different from estimating whether $x$ is more likely to be drawn from some different distribution. Conceptually, for the latter, we would like to compute $p(\text{OOD}|x)$, which by Bayes' rule $p(x|\text{OOD})/p(x)$ up to an $x$-independent constant. In general, knowing $p(x)$ tells us nothing about the value of this ratio. $p(x|\text{OOD})/p(x)$ is also invariant to any coordinate transformation on $x$, whereas $p(x)$ is not.

We illustrate this phenomenon with a simple 1D example in Figure A.10, where the ID data is drawn from $\mathcal{N}(0, 1)$ and the OOD data is drawn from $\mathcal{N}(2, 1)$. Suppose we model the ID data with $x \sim p_\mu(x) = \mathcal{N}(x|\mu, 1)$, where $\mu$ is the parameter of our model. Choosing $\mu = 0$ will exactly model the true distribution and achieve the highest likelihood. However, as shown in Figure A.10 (right), the optimal choice of $\mu$ for OOD detection is $-\infty$, achieving a maximum AUROC but infinite KL divergence from the true ID distribution.

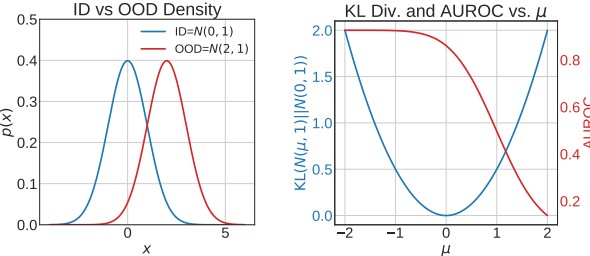

*Figure A.10.* Generative models are not optimal OOD detectors: for generative model $\mathcal{N}(\mu, 1)$, the optimal $\mu$ is 0 to model the ID data but $-\infty$ for OOD detection.

**Measuring typicality rather than density as an alternative for OOD detection** Rather than asking whether a point has a high density, typicality asks whether a point belongs to a region with high probability mass. However, typicality has very similar pathologies compared to density.

One approach of operationalizing typicality is by measuring whether some high-level feature of the data has high likelihood. Consider a common motivating example for typicality: points drawn from a high dimensional Gaussian $\mathcal{N}(0, I)$ in $\mathbb{R}^d$ will have norms $\|x\| = \sqrt{\sum_{i=1}^d x_i^2}$ concentrating around $\sqrt{d}$ by the Law of Large Numbers (LLN). A point at the origin will be considered highly OOD based on typicality, since it has zero norm, yet it has the highest density and will thus be considered highly ID based on the density. But there is no obvious reason why we should judge typicality based on the norm rather than other features of the data. Alternatively, one might consider using the average value of $x$ over the dimensions, $\frac{1}{d} \sum_{i=1}^d x_i$, as the feature, which concentrates around 0 by LLN. Based on this quantity, a point at the origin looks perfectly typical, while a point on a sphere of radius $\sqrt{d}$ looks highly atypical. Therefore, exactly similar to the density, notions of typicality will tend to depend on a subjective choice of how to coarse-grain the input space based on quantities that are most relevant for distinguishing between ID and OOD data. Finally, while the choice of using the norm in this example can be justified via the notion of $(\epsilon, N)$-typical set (Nalisnick et al., 2020), the latter relies again on the density $p(x)$, which is not invariant to coordinate transformations and depends on assumptions on how the data is presented.

**OOD Detection Requires Coarse-Grained Representations.** In general, every test input we encounter will differ from the ID inputs we have previously seen. However, not all test inputs are considered OOD because we are only concerned with certain key differences. When learning a generative model $p(x)$ of the ID data, our goal is not necessarily to capture the distribution of $x$ in its finest details. Instead, for the purpose of OOD detection, it is more appropriate to model the distribution over a coarse-grained representation $h(x)$, which captures the attributes necessary for distinguishing OOD from ID data and ideally nothing more.

Consider an ID dataset consisting of 1000 breeds of dogs and 10 breeds of cats. If our generative model captures the frequency of each individual breed, any dog input we observe will typically be considered 100 times more OOD-like than any cat input based on the likelihood of the generative model. However, if our goal is to detect other animal species and non-animal objects, the likelihood of this model is clearly not aligned with the objective of OOD detection. In this case, it would be more suitable to model only the frequency over the dog and cat categories, which serves as an appropriately

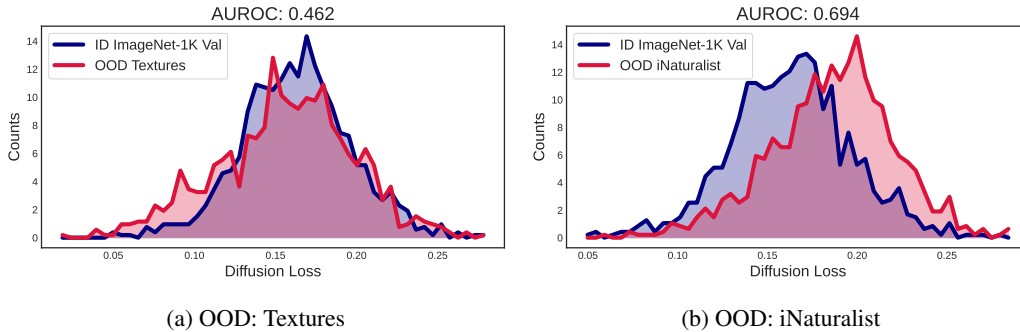

(a) OOD: Textures        (b) OOD: iNaturalist

*Figure A.11.* **Diffusion Models can fail catastrophically at OOD.** (**a**): Using the diffusion loss, the Diffusion Transformer (DiT) (Peebles and Xie, 2023) fails catastrophically at detecting OOD inputs from the Describable Textures dataset. (**b**): the DiT model does decently at detecting OOD inputs from iNaturalist.

coarse-grained representation of the individual breeds. Since the definition of OOD is ultimately user-defined, the correct coarse-grained representation depends on both the dataset and the intended definition of OOD, and it might be challenging to accurately specify even when a definition is known.

In Figure 8 (right), we show the test log-likelihoods (normalized by dimension) of RealNVP (Dinh et al., 2016) normalizing flow models of various sizes trained on CelebA images and their AUROCs for detecting CIFAR-10. While models with the lowest test likelihoods on ID data perform poorly for OOD detection, their OOD detection performance does not improve monotonically with their test likelihoods. In fact, the AUROC eventually decreases with improvements in likelihood.

In Figure 8 (left), we demonstrate the same phenomenon for a feature-space generative model. We construct a Gaussian Mixture Model (GMM) model of the features produced by a ResNet-50 pre-trained on ImageNet-1K, the ID dataset. To optimize for the likelihood on ID data, we choose the cluster means to be the class-conditional means and use the empirical covariance of all features centered by their respective class means as the covariance of the clusters. This GMM model is precisely the generative model used by the Mahalanobis method (Lee et al., 2018). As we interpolate between the empirical covariance and a trivial identity covariance, the ID test likelihood of this GMM model decreases, yet the AUROC for detecting ImageNet-OOD improves monotonically.

**The Impact of Inductive Biases.** How a generative model assigns density to data unseen during training is highly dependent on their inductive biases. Despite being highly flexible density models, normalizing flows are known to be poor OOD detectors when trained as a generative model over raw images because their inductive biases encourage the model to focus on low-level pixel correlations rather than high-level semantic properties (Kirichenko et al., 2020; Nalisnick et al., 2018). Here, we demonstrate that the same failure mode applies to diffusion models, a distinct class of generative models achieving state-of-the-art image generation (Betker et al., 2023; Saharia et al., 2022).

We use the Diffusion Transformer (DiT), a 256x256-resolution latent diffusion model trained on ImageNet-1K (Peebles and Xie, 2023). We score images based on the diffusion loss, a variance-reduced approximation of the variational lower bound (Kingma et al., 2021; Ho et al., 2020). In Figure A.11, we show the DiT fails catastrophically in detecting OOD data from Describable Textures but achieves decent performance in detecting OOD data from iNaturalist.

In Figure A.12, we qualitatively show that a 256x256 DiT trained on ImageNet-1K often accurately reconstructs noised images from Describable Textures despite never having trained on them. We add noise to the inputs corresponding to the diffusion timesteps at 49, 98, 147 out of 249, where higher timesteps are more noisy.

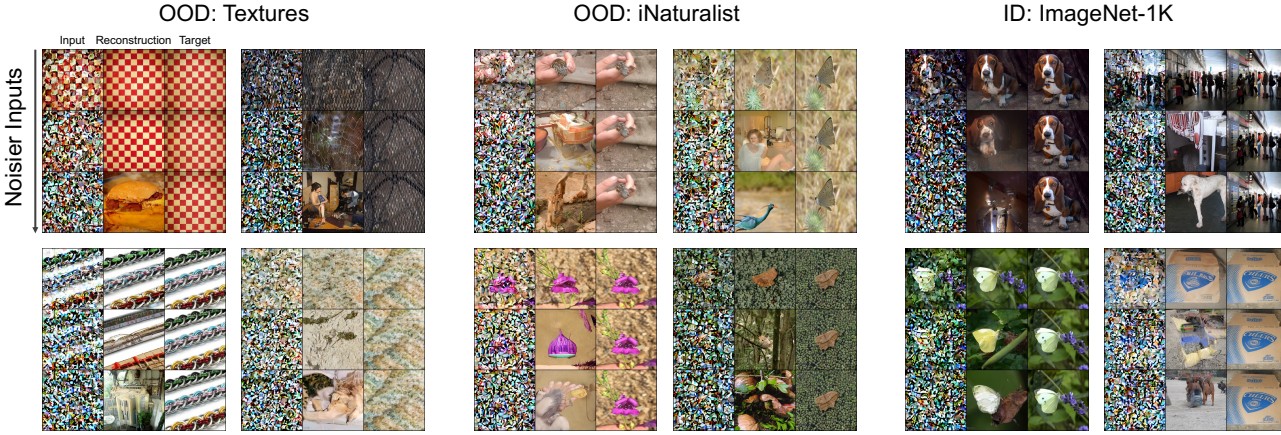

*Figure A.12.* **Visualization of DiT Reconstruction Error.** A DiT trained on ImageNet-1K often accurately reconstructs noised images from Describable Textures despite never having trained on them. **Left**: Reconstructions of noised Describable Textures images compared to **middle**: iNaturalist images and **right**: ImageNet-1K images.

### A.7   Outlier Exposure

Training a model with outlier exposure, where we purposely encourage the model to have high predictive uncertainty as we move away from the training data, is effective for improving OOD detection, and we see that performance is improved for most OOD problems with semantic shifts in Figure A.13 (left). However, this same procedure can significantly hurt the models' ability to generalize under covariate shifts, which is important for model robustness.

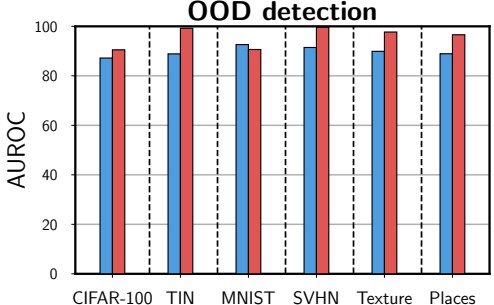 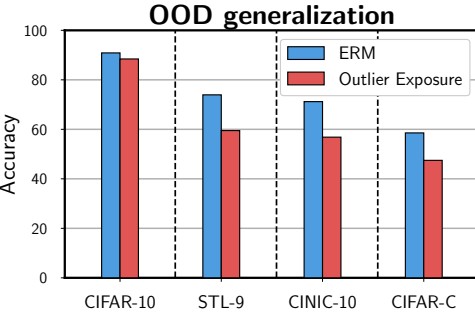

*Figure A.13.* Training a ResNet-18 with outlier exposure improves OOD detection for semantic shift datasets but hurts OOD generalization over covariate shifts.

# B    Implementation details

The code to reproduce our experiments can be found at `https://github.com/yucenli/ood-pathologies`.

## B.1    Outlier exposure experiment.

On Figure 5, we compare OE model to the baseline ERM training in OOD detection (left panel) and OOD generalization (right panel). For semantic shift detection, we use CIFAR-100, Tiny ImageNet, MNIST, SVHN, Textures (Cimpoi et al., 2014), and Places365 (Zhou et al., 2014). For OOD generalization we evaluate on STL-10 (Coates et al., 2011), CINIC-10 (Darlow et al., 2018) and CIFAR-10-C (Hendrycks and Dietterich, 2019).

We adapt OpenOOD codebase (Zhang et al., 2023; Yang et al., 2022) to train ResNet-18 with baseline ERM training and Outlier Exposure (Hendrycks et al., 2018) and evaluate models on OOD detection. We train models for 100 epochs with batch size 128 for ID data and batch size 256 for the outlier dataset, SGD with momentum and initial learning rate 0.1 and weight decay $5 \times 10^{-4}$, and we set the coefficient before the OE loss to $\alpha = 0.5$ (overall, we use standard training hyper-parameters as in Zhang et al. (2023)). For Figure 5, we run both methods with 3 random seeds and report the average performance. To evaluate the model on STL-10, we only use the 9 classes which overlap with CIFAR-10 classes and drop the class "monkey" not present in CIFAR-10 (thus, the evaluation is marked as STL-9 in Figure 5). For CIFAR-C, we report the average accuracy across 15 corruptions (Gaussian Noise, Shot Noise, Impulse Noise, Defocus Blur, Glass Blur, Motion Blur, Zoom Blur, Snow, Frost, Fog, Brightness, Contrast, Elastic transform, Pixelate, JPEG Compression).

## B.2    Evaluating pre-trained models.

We evaluate 54 models from the `timm` and `torchvision` libraries, including 9 different architecture types: ResNet, TinyNet, VGG, MobileNet, ConvNeXt, RegNetY, ReXNet, MLP-Mixer, and ViT; and 6 different pre-training data setups: training on IN-1K from scratch, pre-training on IN-21K and fine-tuning on 1N-1K, pre-training on IN-12K (a subset of IN-21K) and fine-tuning on IN-1K, CLIP (Radford et al., 2021) pre-training on LAION and fine-tuning on IN-1K, CLIP pre-trainig on LAION and further fine-tuning on IN-21K and then IN-1K, and Instagram-1B pre-training and further IN-1K fine-tuning of SEER models (Goyal et al., 2021).

## B.3    Scaling Experiments

We benchmark the following models to demonstrate impact of scale in Figure 9:

1. ResNet-18 trained on ImageNet-1k

2. ResNet-34 trained on ImageNet-1k

3. ResNet-50 trained on ImageNet-1k

4. ViT-S/16 trained on ImageNet-1k

5. ViT-B/16 trained on ImageNet-1k

6. ViT-S/16 trained on ImageNet-1k with DINO

7. ViT-B/16 trained on ImageNet-1k with DINO

8. ViT-B/16 trained with CLIP

9. ViT-L/14 trained with CLIP

10. ViT-B/16 pretrained on CLIP, finetuned on ImageNet-1k

11. ViT-B/14 trained on 142M images with DINOv2

12. ViT-G/14 trained on 142M images with DINOv2

