# OpenReview forum: "Position: Supervised Classifiers Answer the Wrong Questions for OOD Detection"
_ICML.cc/2025/Position_Paper_Track — ICML 2025 Position Paper Track poster_

### Official Review · Reviewer_WGax · 2025-02-25

**Significance:** 4
**Argument Clarity:** 3
**Rating:** 4
**Confidence:** 5

**Questions:**

The paper seems to revolve largely around supervised learning only and moreso classification. How do the arguments hold up when going to different supervised learning tasks?

**Discussion Potential:**

4

**Paper Summary:**

The paper argues that out-of-distribution (OOD) detection in supervised learning is misguided, as the real questions it seems to ask are whether atypical representations manifest or whether unconfident predictions arise. Given the nature of how features are learned in supervised learning, the paper dissects failure modes of popular approaches to OOD detection and supports them through easy experiments.

### POST REBUTTAL UPDATE

As indicated in the comment below, the rebuttal has provided good answers to my questions. I choose to retain my rating and vote to accept the paper.

**Position:**

Yes

**Position In Title:**

Yes

**Related Work:**

3

**Strengths And Weaknesses:**

Strengths
* The discussion of the failure modes is well done and becomes particularly clear through the supportive empirical examples. These experiments make the pathologies very intuitive.
* The paper is very well written and the claims are generally well supported. The first half is intuitively structured and the clear visualizations make it easy to follow the line of argumentation.
* I genuinely enjoyed the paper. Although many of the points have certainly been made in parts, I believe the arguments are sufficiently scattered across the literature that it is worth to aggregate them all in one place. The paper excels in this aspect, and there are few dimensions (see weaknesses) that I would find lacking in the discussion. As such, I find the paper to be both very important and timely, in particular as there is an increasing amount of works that keep putting out algorithms that are likely answering the wrong question.

Weaknesses
* The main weakness and my main concern with the paper is the way the alternative views section is framed. It seems that there is no “real” alternate view here, but rather there are a few sentences that actually fall more in line with what was nicely pointed out in section 4. The section then goes into detail on OOD detection through inclusion of an OOD class, scaling, and combining methods. These are really just more subsections to what has been discussed in section 4 already. They are very interesting and contain important discussions, but they do not really qualify as an alternate view. For instance, saying that scaling models allows to more easily distinguish OOD from ID data, doesn’t really circumvent the pathologies of section 4 as it doesn’t change the paradigm. I think section 5 should really be reframed to a follow-up section on 4 with more nuanced discussion of pitfalls and caveats, rather than being presented as an alternative.
* The only “alternate” view I thought was convincing was 5.6, which could be seen as presenting the alternative position that “OOD detection methods can answer the right questions, but supervised learning limits them fundamentally”. Given the way the introduction was written, I was expecting something along these lines as the alternate position. It is a bit unfortunate that the discussion here is quite short and a bit shallow, with details deferred largely to the appendix. There have been works that have looked into these aspects in more detail, e.g. “Open Set Recognition Through Deep Neural Network Uncertainty: Does Out-of-Distribution Detection Require Generative Classifiers?” Comes to mind, that have at least shown some empirical evidence for p(x) being valuable when used to model the joint likelihood p(x,y) and inferring classes through p(y|x)p(x). I personally agree with the authors that there are still several pathologies and new sets of problems here, but there likely is a much deeper discussion to have related to the fact that supervised learning is even more misaligned with OOD detection because we are not even learning the features to describe the data (as pointed out in the paper multiple times). Intuitively, using generative approaches, even if the goal is classification, thus sounds like a good alternate view that I feel deserves much more in-depth discussion. It would be nice to discuss how moving from supervised to unsupervised or self-supervised learning affects the position put forward.
* There is almost a second position point at the start of the introduction, implying that the correct approach may be about learning invariances. This was rather irritating and somewhat distracting. Apart from the fact that this seems to be a tangential point from the paper’s main premise, it also comes with a completely new set of pathologies that make the “solution” questionable: how do we know all invariances in advance before having observed unknown examples, if there are too many invariances, how can we distinguish cases where they suddenly may become relevant etc. As this doesn’t seem to have much to do with the rest of the paper I don’t find it very detrimental, but I would suggest removing this part in the introduction.
* As the paper seems to revolve primarily around supervised learning and classification, I would suggest to make this clear more strongly in the title and position.

Despite the listed weaknesses, I remain very much in favor of accepting the paper, as I find the argumentation mostly cohesive, the support for the position convincing, and the topic important. The paper manages to convey the arguments clearly and the advantages outweigh the current weakpoints. I would nevertheless suggest to take into account the two above main concerns surrounding the framing of “alternate views” (which isn’t really all that alternate) and generative vs. discriminative modeling.

**Support:**

4

---

> ### Author Rebuttal · Authors · 2025-04-01
>
> Thank you for your review, and we are glad you enjoyed the paper!
>
> **Alternative view section**  Thank you for your feedback! We want to clarify that our paper’s position is that “the entire premise of using supervised models trained on in-distribution data for out-of-distribution detection is fundamentally flawed.” Therefore, when discussing alternative views, defined by the ICML Call for Position Papers as “positions that are opposed to the paper’s position”, we focus on viewpoints which support the counterclaim that “the entire premise of using supervised models trained on in-distribution data for out-of-distribution detection is *not* fundamentally flawed if we [scale model and data size, combine feature and logit-based methods, etc]”.
>
> Each of the opposing views we discuss in Section 5 are supported by existing papers on OOD detection. For example, Wang et al [1] state that the performances of logit-based and feature-based methods “are limited by the singleness of their information source” and thus propose a novel hybrid approach, implying that using supervised models for OOD detection is sufficient as long as both feature and logit-based methods are combined. We will clarify our position and alternative positions in the camera-ready iteration of the paper. Based on your feedback, we are also happy to restructure the paper to better distinguish between methods which directly use supervised models (such as hybrid models and epistemic uncertainty), and methods which incorporate additional techniques for OOD detection (such as pretraining, adding an OOD class, etc).
>
> **Generative models** Although our position focuses on supervised methods, we discuss generative models because they seem like a natural solution to OOD detection. In Appendix A.6, we demonstrate conceptual and practical limitations of generative models, identifying failure modes in normalizing flows as well as diffusion transformers. We will expand on this discussion, and move the detailed discussion on generative modeling approach for OOD detection and its pathologies from Appendix A.6 to the main text.
>
> We hope this has addressed your questions, and we are happy to engage with any further comments!
>
>
> References\
> [1] Wang et al, ViM: Out-Of-Distribution with Virtual-logit Matching\
> [2] D’ Angelo et al, Repulsive Deep Ensembles are Bayesian

---

> > ### Comment · Reviewer_WGax · 2025-04-02
> >
> > I appreciate the rebuttal comments. The clarification regarding the alternative view section is helpful, even if I had likely structured it differently. But this is okay and remains a choice. The point on generative models is also appreciated.
> >
> > Overall, I remain happy with the paper and choose to retain my rating. I believe the paper should be accepted.

---

### Official Review · Reviewer_mAah · 2025-03-07

**Significance:** 3
**Argument Clarity:** 3
**Rating:** 4
**Confidence:** 4

**Questions:**

- L201L: what does "addition of an Oracle PCA projection (in blue)" mean? Is the 10% increment supposed to be the contribution from indistinguishable features? It is a bit hard for me to follow Figure 2 left and the procedure to generate it. I understand the high level concept that PCA projections on highly discriminating dimensions is supposed to give you higher performance. But not hundred percent sure what that text is saying.
- I am not sure I understand the epistemic uncertainty part completely. Considering the infinite ID data case, the epistemic uncertainty only goes to zero for ID samples. OOD samples may still be unlikely for the distribution learned, resulting in high epistemic uncertainty of the model over those samples. So if I am misunderstanding this: "If measuring epistemic uncertainty were the correct approach to OOD detection, then such low epistemic uncertainty implies that OOD points do not exist in this setting. Therefore, because perfectly capturing epistemic uncertainty is not enough to solve OOD detection, they must answer fundamentally different questions", please explain this more simply. Would also like to see details of the experiment done perhaps in the appendix. The cited reference is a book, maybe a simpler explanation would help the reader.
- Would maybe recommend moving the real generative model experiments to the main text and the 1D example case to the appendix?

**Discussion Potential:**

3

**Paper Summary:**

This paper presents a novel position against the existing literature of OOD detection methods, that methods which only use in-distribution data for OOD detection are fundamentally flawed. They show that it may be impossible to do this in the general case because both feature and logit-based methods can fail to capture the information or relationship which is necessary for detecting some class of OOD samples. They argue that if the aim is to detect OOD samples, the current methods are incomplete. And if the aim is to just perform well under well-specified shifts (like covariate shift), OOD detection based interventions can be more harmful than useful.

## Update after rebuttal.
Authors answered my concerns as well as they could and I retained my original rating of Accept.

**Position:**

Yes

**Position In Title:**

Yes

**Related Work:**

3

**Strengths And Weaknesses:**

Strengths:
- The experiment design for all experiments is deliberate, pretty exhaustive and overall great. These experiments convey a lot of useful information supporting the hypothesis.
- The paper is extremely easy to read and follow. The flow is excellent. Figures and presentation are thought provoking and remarkably to the point. Very well written paper.

Weaknesses:
- At several places the wording is stronger than it should be, based on the evidence provided. For example, L131R: "it is impossible to infer these most discriminating features without access to OOD data". This statement needs supporting theory. An example in Figure 1 shows that a model that only learns one kind of feature, will find it impossible to detect OOD samples based on other features. But this can not be assumed of all models learned with supervised learning on ID classes. There exist models that can do ID classification and learn these OOD discriminating features, both at the same time. These models may be hard to find, but to say impossible is to say these models do not exist which needs more support than provided. A more reasonable statement may be, "in practice, models typically do not find the most discriminating features without access to OOD data". Similarly: "it is generally impossible to distinguish the features that discriminate between OOD and ID data from the features that are irrelevant without access to OOD data", "2) requires information unavailable to any feature-based method", "This result shows that, as long as the OOD dataset is not specified at training time, removing the influence of irrelevant features is impossible for any feature-based method", "This inherit misalignment of goals means no logit-based methods can overcome this pathology."etc. are strong statements made without proper supporting justification.
- This is also a drawback of this study: it conflates what happens with common models trained on ID datasets, as the universal case for all models and assumes that the problem is unsolvable. For models that are trained to distinguish between trucks and automobiles, it is not surprising for it to classify OOD inputs confidently in one of the two domains, because the original classification problem is itself under-specified. Therefore, some more emphasis on the origin of these problems and potential solutions in practical models would have been helpful instead of generalizing it as a universal problem.
- The discussion around natural ID uncertainty and OOD uncertainty seems to be a little confusing. Natural ID uncertainty is clear as it the uncertainty of being classified into one of the ID classes given the information about those classes through training samples. Figure 3 (a) also presents this excellently. But natural OOD uncertainty is not clear. It is true that the OOD certainty would depend on the type of ID data and the information it contains. Some OOD examples may naturally have low uncertainty in mild-shifts (like detecting cars in adverse weather), but naturally high uncertainty in extreme shifts (like detecting cars for a dog vs cat classifier).


Overall, I generally agree with most of the paper's content and arguments. OOD detection and its usage in avoiding incorrect test-time predictions is often under-specified without advance knowledge of the type of shift expected. The findings on small to large models on a wide variety of datasets through appropriate experiment design is very convincing. I support the paper for acceptance at this venue. But I would mention that a lot of recent literature attempts to solve specific problems like trustworthy prediction under covariate shift. A good specification of the types of problems being aimed can be seen in Figure 2 of [this paper](https://arxiv.org/pdf/2110.11334). This maybe a good way to categorize the problem before solving it, and not attempting a general solution which may not exist. Lastly, I would still want to see the paper improve on its strong phrasing of conclusions which may not be properly supported, and will look for an update to the draft regarding the same. Another thing I found missing in this paper is an alternate position, or how would the authors go about solving the highlighted issues? I don't expect well formed solutions, but some potential directions in the discussion section would be good.
Good job overall!

**Support:**

4

---

> ### Author Rebuttal · Authors · 2025-04-01
>
> Thank you for your positive review and feedback!
>
> **Softening the claims.** Thank you for your feedback! We will update our wording to reflect our position more clearly.
>
> **ID and OOD uncertainty.** We would like to clarify that we do not make any claims on what the uncertainty of OOD examples “naturally” should be: instead, we note that uncertainty-based OOD detection methods rely on the incorrect assumption that the uncertainty of OOD examples should be **different from the uncertainty of ID examples**. For instance, in the example of detecting OOD dogs for a car vs truck classifier (Figure A.5), we find that dog inputs have very low label uncertainty despite representing an extreme distribution shift. We do not claim that this behavior is innately correct or incorrect, but we do identify that this **low uncertainty causes uncertainty-based OOD detection methods to fail.**
>
> **Prior works on trustworthy predictions.** We will add a discussion of Yang et al “Generalized OOD detection“; thank you for providing the reference! We would like to emphasize that in our paper we consider the OOD detection problem, as you mentioned, without the advance knowledge of the type of shift expected, and thus, in the categorization of Figure 2 of Yang et al we consider cases (a)-(d). As you pointed out, the general solution for these cases relying on the supervised classifier is challenging due to misspecifications we describe in the paper.
>
> **Figure 2.** The “irrelevant features” error represented in blue in Figure 2 represents the difference in AUROC between using Mahalanobis distance on all features compared to using Mahalanobis distance using only the most relevant features for the specific OOD detection task. We identify this set of features through an oracle PCA projection, which finds the dimensions which best differentiate ID from OOD data. Therefore, the blue represents the gap in OOD detection performance which is due to using less relevant features for the Mahalanobis distance computation. We will add this explanation to our final manuscript for clarity.
>
> **Epistemic uncertainty.** The epistemic uncertainty in our work refers to the uncertainty over model parameters rather than predictive uncertainty. We place a prior distribution over the model weights $\Theta \sim q(\theta)$, and we evaluate the posterior $p(\Theta \mid X,Y)$ (see Section 2.1 of [1] for further details). As the model observes more training data, the posterior concentrates, reducing the uncertainty over model parameters for any input, whether ID or OOD. We demonstrate this behavior in Figure 7, where we show that increasing the amount of ID data leads to decreased uncertainty over model parameters for both ID and OOD inputs. Therefore, this experiment shows that measuring the epistemic uncertainty over parameters is fundamentally different from answering OOD detection. We will update the text with additional references and clarifications.
>
> **Generative models.** We will move up the generative model experiment to the main text in the next revision, thank you for the suggestion!
>
> We hope this has addressed your questions, and we are happy to engage with any further comments!
>
> [1] Kendall et al, What Uncertainties Do We Need in Bayesian Deep Learning for Computer Vision?

---

> > ### Comment · Reviewer_mAah · 2025-04-02
> >
> > Thanks for the clarifications.
> >
> > About epistemic uncertainty, I see the point regarding uncertainty over parameters instead of predictions. What do you say about methods which model uncertainty over predictions? In relation to OOD detection.

---

> > > ### Author Response · Authors · 2025-04-03
> > >
> > > This is a great question! The epistemic uncertainty over predictions is actually manifested through the
> > > epistemic uncertainty over parameters. This is visualized in Figure 7: as we increase the number of data points, the epistemic uncertainty over parameters decreases, and thus the epistemic uncertainty in the prediction space also decreases, as indicated by the more concentrated regions of low MSP. Therefore, the same arguments for epistemic uncertainty over parameters would also hold for epistemic uncertainty over predictions.
> > >
> > > The situation with epistemic uncertainty over parameters is particularly interesting, because for small amounts of in-distribution data, we do expect it to help with OOD detection, because it ends up leading to predictive uncertainty as we move away from the ID data manifold. But because the posterior over parameters (and thus decision boundaries) collapses in the limit of ID data, these methods end up getting worse at detecting OOD data, the more ID data we have seen, which is the opposite of the behavior we would hope for! In principle, with a well-specified procedure, it should get easier to detect OOD points if we see more ID points. Thanks for your question.

---

### Official Review · Reviewer_HesL · 2025-03-13

**Significance:** 4
**Argument Clarity:** 4
**Rating:** 4
**Confidence:** 5

**Questions:**

None at the moment.

**Discussion Potential:**

4

**Paper Summary:**

The position is that the current literature and methods on "OOD" detection are misaligned and asking the wrong questions.  The paper argues that using supervised models trained on in-distribution data is flawed. The position is that a dog (unseen class) should be distinguishable from trucks and cars (seen classes) but not by models that have been trained to distinguish between trucks and cars.  "Supervised models can only determine if an input leads to atypical representations or uncertain predictions,which is fundamentally different than determining if the input belongs to the training distribution." (Fig 1 caption) summarizes the position.

**Position:**

Yes

**Position In Title:**

Yes

**Related Work:**

3

**Strengths And Weaknesses:**

**Strengths:**
1. Page 1 states the position nicely: "It is not that OOD detection is “fundamentally difficult,” but rather that detection is being approached with methods that answer the wrong question—a dog on the whole should be distinguishable from trucks and cars, but not by training a supervised model to only differentiate between trucks and cars".  I think this position is important as what the current paradigm of OOD detction is doing is to comment on whether or not the classifier has uncertainty (logits closer to uniform) and whether or not features are atypical, but not whether the test input is OOD.  In some sense this is a comment on the choice of doing indirect OOD detection via features / logits when these are different problems.
2. I think the paper does an excellent job at highlighting the key aspects of the position.  Comment 1 contains one of those but others such as "OOD features can sometimes be indistinguishable from ID features", "ID examples often have high uncertainty / OOD examples often have low uncertainty" etc are equally important.
3. Paper is well written and there is good focus on the stated position.  The stated position itself is of importance to the ML community and the position is provocative to fuel discussion in the community.

**Weaknesses:**
1. The focus is only on image classification experiments.  Perhaps a set of experiments in other domains could help, although I understand that existing literature is also focused on image classification experiments.

**Support:**

4

---

> ### Author Rebuttal · Authors · 2025-04-01
>
> Thank you for your review, and we are glad that you think that our position is “of importance to the ML community”!
>
> In response to your feedback, we have incorporated additional experiments over the language domain. Specifically, we consider the Multi-NLI dataset [1] as in-distribution, where the dataset consists of a premise and a hypothesis and classifies their relationship as entailment, contradiction, or neutral. Multi-NLI covers a wide range of genres such as fiction and conversation transcripts. We consider two different OOD datasets for detecting distribution shifts: the SNLI dataset [2], which consists solely of data drawn from image captions, and the QNLI dataset [3], which is adapted from a question answering dataset from Wikipedia. We benchmark the OOD detection performance of BART [4] trained on the Multi-NLI dataset (“facebook/bart-large-mnli” on Hugging Face), which achieves a 0.963 accuracy on ID test data. We report the following FPR@95 scores (false positive rate when true positive rate is 0.95) for logit-based methods:
>
> | OOD Dataset | MSP   | Max-Logit | Entropy | Energy Score |
> |-------------|-------|-----------|---------|--------------|
> | SNLI        | 0.956 | 0.950     | 0.957   | 0.950        |
> | QNLI        | 0.826 | 0.884     | 0.827   | 0.883        |
>
> We find that the supervised model which was only trained on Multi-NLI is consistently over-confident on OOD data, and a majority of OOD samples are incorrectly classified as ID. Due to the limited rebuttal period, we only provide results for one model; however, we will cover a wider range of models to demonstrate the generality of our claims for the camera-ready version.
>
> We hope this has addressed your questions, and we are happy to engage with any further comments.
>
> [1] Williams et al, A Broad-Coverage Challenge Corpus for Sentence Understanding through Inference\
> [2] Bowman et al, A Large Annotated Corpus for Learning Natural Language Inference\
> [3] Wang et al, GLUE: A Multi-Task Benchmark and Analysis Platform for Natural Language Understanding\
> [4] Lewis et al, BART: Denoising Sequence-to-Sequence Pre-training for Natural Language Generation, Translation, and Comprehension

---

> > ### Comment · Reviewer_HesL · 2025-04-03
> >
> > Thanks for the response and providing NLP experiments -- not surprised to see models being overconfident on OOD data (and since most detection techniques are based on confidence/energy -- that will naturally lead to " a majority of OOD samples are incorrectly classified as ID" like you observe.  Great that you're running more experiments for camera ready -- that will further make this point clear.  There are countless papers on OOD detection chasing benchmarks -- but if those benchmarks themselves are flawed (or at least not measuring what they're supposed to measure) we need to communicate that.
> >
> > I am recommending acceptance, as are 2 other reviewers.  I have no further questions.

---

### Official Review · Reviewer_sDfa · 2025-03-14

**Significance:** 2
**Argument Clarity:** 2
**Rating:** 1
**Confidence:** 4

**Questions:**

None.

**Discussion Potential:**

2

**Paper Summary:**

This paper addresses the problem of out-of-distribution (OOD) detection and argues that the two popular families of OOD detection methods (features-based and logins-based) are fundamentally answering the wrong question. The authors have undertaken some analyses to illustrate why these methods are ineffective for OOD detection.

**Position:**

Yes

**Position In Title:**

Yes

**Related Work:**

1

**Strengths And Weaknesses:**

Strengths:
- This paper is well-written and easy to understand.
- The use of illustrative examples effectively conveys its arguments and makes the discussions more effective.

Weaknesses:
- This paper lacks a clear formulation of the OOD detection problem. Based on its analyses, it appears to frame the task as simultaneously learning a predictive model for in-distribution classification and OOD detection. However, this formulation may not be suited to address the binary classification problem of OOD detection. In my view, this may be a key reason why many methods in this category are ineffective for OOD detection.
- The scope of this paper is somehow narrow. Its position is primarily based on examining supervised OOD detection methods for image classification, while overlooking the family of unsupervised OOD detection methods, in particular density estimation methods, as well as the related anomaly detection methods.
- This paper does not provide significant new insights into alternative perspectives of tackling the OOD detection problems more effectively. Many of its findings have already been discussed in the existing related studies.

**Support:**

3

---

> ### Author Rebuttal · Authors · 2025-04-01
>
> Thank you for your review. We would like to clarify a few important misunderstandings.
>
> **Scope of the paper** Our paper addresses the popular family of methods which take a pre-existing model trained on a supervised classification task and use the features or logits of this model to perform OOD detection. The supervised task and OOD detection are not learned simultaneously; instead, we focus on the setting where the model is trained only to perform in-distribution classification and then expected to perform OOD detection. This is an extremely common approach to OOD detection and encompasses a broad line of research [1,2,3,4,5,6,7,8, etc]. Although our position paper is focused on methods which rely on supervised models, we also identify the limitations of unsupervised OOD detection methods like generative modeling in Section 5.6.
>
> **Position of the paper.** We fully agree that learning a predictive model for in-distribution classification is not a formulation that is suited to properly address OOD detection. In fact, this is the position that we argue for in the paper, and this position is in direct opposition to much existing work: many SoTA OOD detection methods focus on adaptations within this existing framework, often implying the opposing stance that improved methodology can address these innate limitations. For example, Sun et al [1] attribute the problem with feature-based approaches to the distributional parameterization and thus propose a feature-based approach. Wei et al [2] attribute the problem with logit-based approaches to model overconfidence and thus propose a novel logit-based method. Wang et al [3] state that the performances of logit-based and feature-based methods “are limited by the singleness of their information source” and thus propose a novel hybrid approach.
>
> In contrast, we argue that these novel methods, or any other logit-based or feature-based methods, still have **fundamental limitations that arise from the misspecification** of supervised training. Although there have been many works which criticize specific OOD detection procedures, no prior work has attributed these limitations to the inherent pathologies of OOD detection methods which cannot be mitigated by improved methodology.
>
> **Novel contributions of our position paper.** First, we would like to emphasize that according to ICML position track: “Submissions to the main ICML conference track emphasize original research and novel results. In contrast, submissions to the position paper track are judged primarily on whether they present a compelling position that warrants greater exposure within the machine learning community…” The goal of this track is to highlight papers that stimulate … discussion on timely topics that need our community’s input.” (https://icml.cc/Conferences/2025/CallForPositionPapers)
> The topic and the position we present is significant and timely (see reviewers A, B, C), and contradicts the arguments expressed in the line of methods being proposed for OOD detection.
>
> While it should not have a significant bearing on the evaluation of our paper, given the ICML guidance on evaluation of position track papers, we do nonetheless make several key novel contributions which demonstrate the limitations of OOD detection methods. We demonstrate the existence of irreducible error for all feature-space methods regardless of methodology, which was previously unexplored and unquantified. We illustrate that this error does not disappear even as we increase model size and data size, disproving the effectiveness of a commonly proposed fix for OOD detection. Similarly, we identify sources of irreducible error for logit-based methods, even at scale. We further illuminate the pathologies of potential remedies such as outlier exposure, Bayesian inference, introducing an OOD class, and generative modeling. Our paper provides the important contribution of understanding the extent to which these pathologies occur, even for large models trained on Internet-scale data.
>
> Finally, we note that all three other reviewers solidly endorse accepting the paper, highlighting the clarity and strength of our position, and the value of bringing all of our observations in one place.
>
>
> [1] Sun et al, Out-of-Distribution Detection with Deep Nearest Neighbors\
> [2] Wei et al, Mitigating Neural Network Overconfidence with Logit Normalization\
> [3] Wang et al, ViM: Out-Of-Distribution with Virtual-logit Matching\
> [4] Hendrycks et al, A Baseline for Detecting Misclassified and Out-of-Distribution Examples in Neural Networks\
> [5] Lee et al, A Simple Unified Framework for Detecting Out-of-Distribution Samples and Adversarial attacks\
> [6] Liang et al, Enhancing the Reliability of Out-of-Distribution Image Detection in Neural Networks\
> [7] Malinin et al, Ensemble Distribution Distillation\
> [8] Tran et al, Plex: Towards Reliability Using Pretrained Large Model Extensions

---

### Decision · Program_Chairs · 2025-04-27

**Decision:**

Accept (poster)

**Comment:**

We have updated the decision on this paper due to the favorable reviews and further discussion amongst reviewers and area chair that provide persuasive rebuttals to the dissenting reviewer's concerns.  In addition to other revisions, the authors should revise the title to clearly communicate the scope of the position (not all OOD methods but just the ones that depend on models trained for supervised classification).

---
Original meta-review:

Several reviewers pointed out that the raised position that supervised learning cannot be used for OOD detection is valid but somewhat narrow in scope. The discussion on unsupervised method for OOD detection is quite limited.

I would recommend to reject this paper.